



**Easy-to-use spatial Random Forest-based downscaling-calibration method for**
**producing high resolution and accurate precipitation data**
Chuanfa Chen[1,2], Baojian Hu[1,2], Yanyan Li[1,2*]
[1] College of Geodesy and Geomatics, Shandong University of Science and
Technology, Qingdao 266590, China
[2] Key Laboratory of Geomatics and Digital Technology of Shandong Province,
Shandong University of Science and Technology, Qingdao 266590, China
* Correspondence: Yanyan Li (yylee@whu.edu.cn)
**Abstract**
High resolution and accurate precipitation data is significantly important for
numerous hydrological applications. To enhance the spatial resolution and accuracy of
satellite-based precipitation products, an easy-to-use downscaling-calibration method
based on spatial Random Forest (SRF) is proposed in this paper, where the spatial
autocorrelation between precipitation measurements is taken into account. The
proposed method consists of two main stages. Firstly, the satellite-based precipitation
was downscaled by SRF with the incorporation of some high-resolution covariates
including latitude, longitude, DEM, NDVI, terrain slope, aspect, relief, and land
surface temperatures. Then, the downscaled precipitation was calibrated by SRF with
rain gauge observations and the aforementioned high-resolution variables. The
monthly Integrated MultisatellitE Retrievals for Global Precipitation Measurement
(IMERG) located in Sichuan province, China from 2015 to 2019 was processed using
our method and its results were compared with those of some classical methods



including geographically weighted regression (GWR), artificial neural network
(ANN), random forest (RF), kriging interpolation only on gauge measurements,
bilinear interpolation-based downscaling and then SRF-based calibration (Bi-SRF),
and SRF-based downscaling and then geographical difference analysis (GDA)-based
calibration (SRF-GDA). Results show that: (1) the proposed method outperforms the
other methods as well as the original IMERG; (2) the monthly-based SRF estimation
is slightly more accurate than the annual-based SRF fraction disaggregation method;
(3) SRF-based downscaling and calibration preforms better than bilinear downscaling
(Bi-SRF) and GDA-based calibration (SRF-GDA); (4) kriging seems more accurate
than GWR and ANN in terms of quantitative accuracy measures, whereas its
precipitation map cannot capture the detailed spatial precipitation patterns; and (5)
among the predictors for calibration, the precipitation interpolated by kriging on the
gauge measurements is the most important variable, indicating the significance for the
inclusion of spatial autocorrelation information in gauge measurements.
**Keywords**: IMERG; Downscaling; Calibration; Machine learning; Interpolation
**1. Introduction**
Precipitation is an important variable for promoting our understanding of
hydrological cycle and water resource management (Chen et al., 2010). Previous
studies showed that about 70-80% of hydrological modeling errors were caused by
precipitation data uncertainties (Gebregiorgis and Hossain, 2013). However,
precipitation is also the most difficult meteorological factor to estimate due to its high



spatial and temporal heterogeneity (Beck et al., 2019). Although rain gauge
observations are reliable and accurate, it is difficult to reflect the spatial precipitation
pattern with the sparse and uneven distribution and limited coverage, especially in
remote and mountainous areas (Ullah et al., 2020).
During the past decades, plenty of satellite-based precipitation datasets have been
produced at regional, quasi-global and fully global scales, such as the Climate
Hazards Group Infrared Precipitation with Station data (CHIRPS, 0.05°) (Funk et al.,
2015), the Precipitation Estimation from Remotely Sensed Information using
Artificial Neural Networks-Climate Data Record (PERSIANN-CDR, 0.25°) (Ashouri
et al., 2015), the Climate Prediction Center (CPC) morphing technique (CMORPH,
0.25°) (Haile et al., 2013), the Multi-Source Weighted-Ensemble Precipitation
(MSWEP, 0.1°) (Beck et al., 2017), the Tropical Rainfall Measuring Mission (TRMM)
Multi-satellite Precipitation Analysis (TMPA, 0.25°) (Huffman et al., 2007) and the
Integrated MultisatellitE Retrievals for Global Precipitation Measurement (GPM)
mission (IMERG, 0.1°) (Hou et al., 2014). Nevertheless, these products are
characterized by considerable systematic biases due to the shortcomings of retrieval
algorithms, sensor capability and spatiotemporal collection frequency (Chen et al.,
2018; Wu et al., 2018; Yang et al., 2017). Moreover, their resolutions (from 0.05° to
2.5°) are too coarse to describe meso- and micro-scale precipitation patterns for
hydrological studies at local and basin scales (Immerzeel et al., 2009). Hence,
downscaling and calibration with the intention of improving the resolution and quality
of satellite-based precipitation datasets has become an essential step prior to various



hydrological applications at local scales (Bhuiyan et al., 2018).
Downscaling provides an effective way to derive high resolution precipitation
products, which is generally achieved by constructing the relationship between
precipitation and environmental variables at a coarse scale, and then putting the
high-resolution variables into the constructed model to downscale the precipitation
data from the coarse resolution to the fine (Chen et al., 2010; Immerzeel et al., 2009).
At present, many downscaling models have been proposed. For example, Immerzeel
et al. (2009) employed an exponential regression (ER) to describe the relationship
between Tropical Rainfall Measuring Mission (TRMM) and Normalized Difference
Vegetation Index (NDVI). Jia et al. (2011) used a multiple linear regression model
(MLR) to establish the relationship between TRMM, digital elevation model (DEM)
and NDVI. Duan and Bastiaanssen (2013) proposed a downscaling model based on
the second-order polynomial relationship between TRMM and NDVI. Considering
the heterogeneous relationship between precipitation and the land surface variables
across the study areas, geographically weighted regression (GWR) was commonly
adopted (Chen et al., 2015; Chen et al., 2014; Chen et al., 2020c; Li et al., 2019; Lu et
al., 2020; Xu et al., 2015), and showed more accurate results than ER and MLR. In
the recent decade, some data-driven machine learning (ML) methods such as random
forests (RF) (Shi et al., 2015; Zhang et al., 2021), support vector machine (SVM)
(Chen et al., 2010; Jing et al., 2016) and artificial neural network (ANN) (Elnashar et
al., 2020) were employed to capture the complex nonlinear relationship between
precipitation and the predictors. However, the downscaled precipitation products



inevitably contain large systematic biases.
To alleviate the inherent biases, many calibration methods have been proposed for
merging gauge observations and satellite-based precipitation to improve the accuracy
and spatial coverage of precipitation, such as nonparametric kernel smoothing method
(Li and Shao, 2010), geographical difference analysis (GDA) (Cheema and
Bastiaanssen, 2012), geographical ratio analysis (GRA) (Duan and Bastiaanssen,
2013), conditional merging (CM) (Berndt et al., 2014), quantile mapping (Chen et al.,
2013; Zhang and Tang, 2015), optimal interpolation (Lu et al., 2020; Wu et al., 2018;
Xie and Xiong, 2011), GWR (Chao et al., 2018; Chen et al., 2018; Lu et al., 2019) and
geostatistical interpolation (Park et al., 2017). However, these methods are based on
some strict assumptions which might not be satisfied in practice (Wu et al., 2020;
Zhang et al., 2021). Moreover, the precipitation-related environmental variables were
not taken into account. To this end, ML-based calibration methods have become
popular, such as Quantile Regression Forests (QRF) (Bhuiyan et al., 2018), ANN
(Pham et al., 2020; Yang and Luo, 2014), deep neural network (Tao et al., 2016), RF
(Baez-Villanueva et al., 2020), convolutional neural network (CNN) (Wu et al., 2020),
SVM and extreme learning machine (Zhang et al., 2021). In contrast, RF with
excellent results has been widely adopted in plenty of studies (Baez-Villanueva et al.,
2020; Bhuiyan et al., 2020).
In the context of downscaling and calibration of precipitation data, the merits of the
ML-based methods include (Hengl et al., 2018; Zhang et al., 2021): (i) they require no
strict statistical assumptions; (ii) they can capture complex nonlinear relationship



between precipitation and the environmental variables; (iii) they can include various

types of predictors, without suffering from the collinearity problem and (iv) they are

generally more accurate than the classical regression methods. However, there are at

least two limitations: (i) the ML algorithms were simply taken as a statistical tool

without considering the spatial autocorrelation between precipitation measurements;

and (ii) the ML algorithms were adopted in either downscaling or calibration, without

being used in both downscaling and calibration. More specifically, some (Jing et al.,

2016; Karbalaye Ghorbanpour et al., 2021; Yan et al., 2021) attempted to use the ML

methods for downscaling and then use the classical method (e.g. GDA and cokriging)

for calibration, while some (Zhang et al., 2021) employed the classical interpolation

methods (e.g. bilinear interpolation and kriging) for downscaling and then used the

ML methods for calibration. However, we regard that the use of ML methods in both

of downscaling and calibration could further improve the accuracy of precipitation,

since the high resolution environmental variables with valuable information can be

fully used in the two stages. To the best of our knowledge, no previous studies have

used the ML technique in both downscaling and calibration with the consideration of

high resolution environmental variables, simultaneously.

Based on aforementioned discussion, the objectives of this study are twofold: (i) to

develop an easy-to-use spatial RF (SRF) by taking into account the spatial

autocorrelation between adjacent gauge measurements, and (ii) to propose a

downscaling-calibration method based on SRF for producing high resolution and

accurate precipitation data. The use of RF as the basic model in our study is mainly



due to its high interpolation accuracy and low computational cost (Belgiu et al., 2016;
Mohsenzadeh Karimi et al., 2020).

Overall, the proposed method consists of two main steps. First, the precipitation

data is downscaled by SRF with the incorporation of some environmental variables
including DEM, NDVI, land surface temperatures (LSTs), terrain parameters, latitude
and longitude as recommended in previous studies (Jing et al., 2016; Li et al., 2019).
Second, SRF and the environmental variables were further used for merging the
downscaled precipitation data and gauge observations to boost the accuracy of the
precipitation data. The merit of the proposed method is that a new spatial RF is
developed for both downscaling and calibration of precipitation products, with the
incorporation of high-resolution environmental variables.
**2 Study area and dataset**
*2.1. Study area*

Sichuan province between 97°21'-108°31'E and 26°03'-34°19'N was selected as the

study area (Fig. 1). It is situated between the Qinghai-Tibet Plateau and the Plain of
the Middle-and-lower Reaches of Yangtze River, with an area of 486,000 km$^2$.
Sichuan province has a complex and varied topography consisting of mountains, hills,
plain basins and plateaus with the elevation ranging from approximately 180 m in the
east to 7100 m in the west. Due to the different topographies in the west and east, the
climate has a significant difference. The east basin has subtropical monsoon climate.
The weather is generally warm, humid and foggy with much cloud, fog and rain but
less sunshine. Most rain gathers from July to September, accounting for 80% of total
annual precipitation. While in the west plateau, the weather is relatively cool or cold.
The climate is featured by a long cold winter, a very short summer and rich sunshine
but less rainfall. Thus, annual precipitation shows significant spatial heterogeneity,
varying from about 400 mm in the west to 1800 mm in the east and with the average
annual precipitation of about 1000 mm. Overall, the high spatial and temporal
variability of precipitation with the complex topography makes the study site ideally
suitable for the evaluation of satellite-based precipitation estimates.

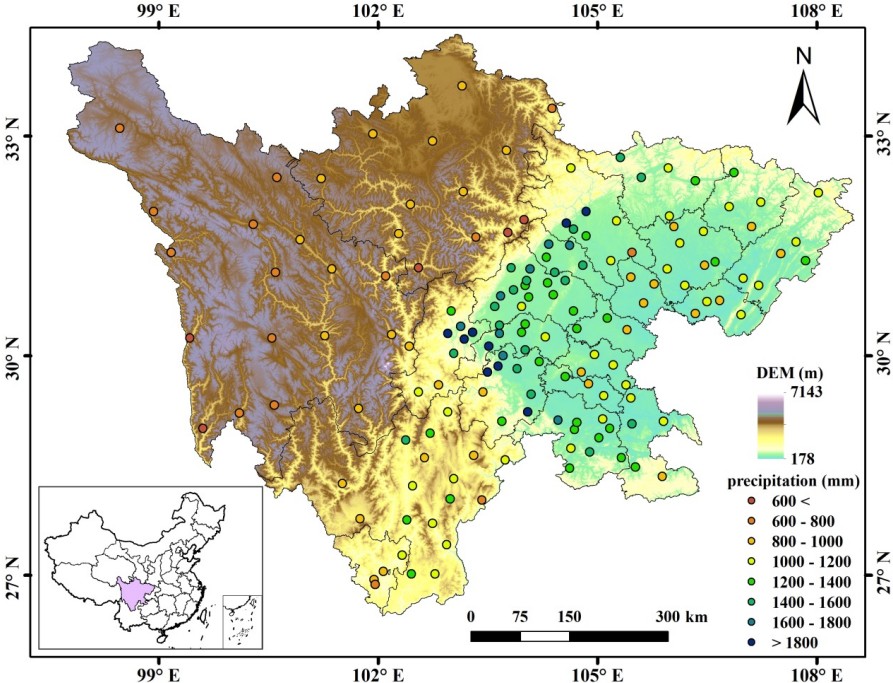


Fig. 1 Topography, distribution of rain gauges and geographic location of Sichuan
province in China
*2.2. Dataset*



### 2.2.1. Rain gauge observations

The study region has 156 rain gauge stations, which shows an unevenly distribution with high density in the east and low density in the west (Fig. 1). On average, the cover area of one rain gauge observation is about 3115 km$^2$. Daily precipitation data from all the stations for the period 2015–2019 were collected from the China Meteorological Data Service Center (CMDSC, http://data.cma.cn/). The data quality was guaranteed based on some strict quality controls, such as manual inspection, outlier check and spatiotemporal consistency verification (Zhao and Yatagai, 2014). After that, the monthly precipitation was produced by aggregating the daily precipitation of rain gauges for each month.

### 2.2.2. Integrated MultisatellitE Retrievals for Global Precipitation Measurement (IMERG)

As the successor of TRMM, the National Aeronautics and Space Administration (NASA) and the Japan Aerospace Exploration Agency (JAXA) initiated the next-generation global precipitation observation mission (Hou et al., 2014). The IMERG products were produced by assimilating all microwave and infrared (IR) estimates, together with gauge observations (Huffman et al., 2019). It has the spatial resolution of 0.1° × 0.1° with the coverage from 60°S-60°N. IMERG provides three different products including Early, Late, and Final Runs, which were computed about 4 hours, 14 hours, and 3.5 months after observation time, respectively. Due to the incorporation of the Global Precipitation Climatology Centre (GPCC) rain gauge data, IMERG Final Run is more accurate than the others (Lu et al., 2019). Thus, the



monthly IMERG V06B Final Run product was adopted in the study. It was
downloaded from https://gpm.nasa.gov/data.
The mean monthly precipitations based on all rain gauges and IMERG during
2015-2019 are shown in Fig. 2. Obviously, IMERG has an overestimation in most
months and the wettest month is July 2018.

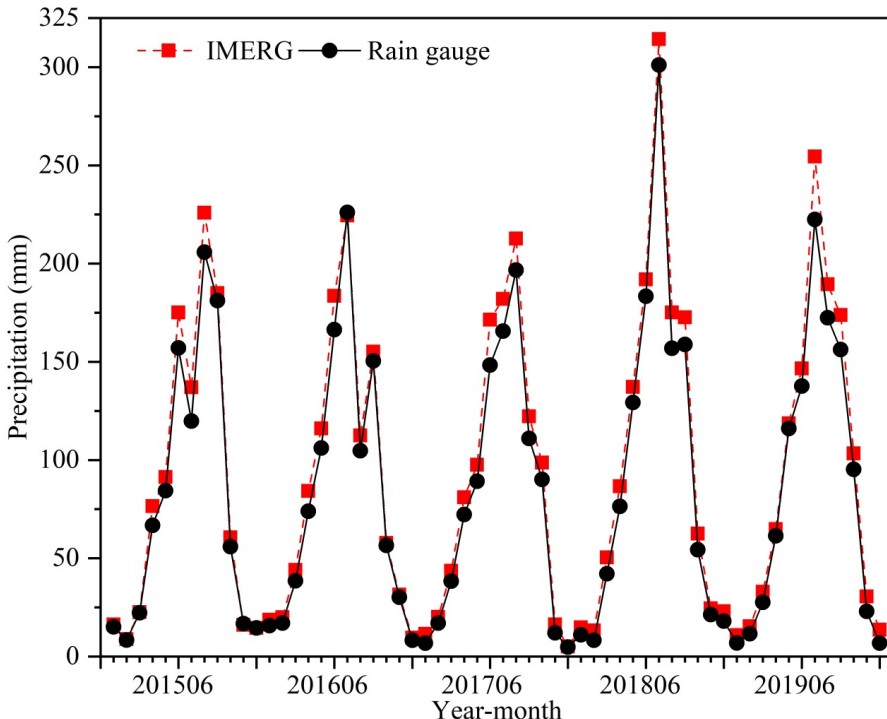


Fig. 2 Mean monthly precipitation based on rain gauges and IMERG from 2015-2019

over Sichuan province

2.2.3. Environmental variables
The Moderate Resolution Imaging Spectroradiometer (MODIS) onboard the
NASA's Terra and Aqua platforms provides plenty of products in global dynamics,
oceans and land processes. The MODIS monthly NDVI with the resolution of 1 km



199 (MOD13A3) from 2015 to 2019 was used in the study and downloaded from

200 International Scientific and Technical Data Mirror Site, Computer Network

201 Information Center of the Chinese Academy of Sciences (http://www.gscloud.cn/).

202 MODIS 8-day LST with the resolution of 1 km (MOD11A2) from 2015 to 2019 was

203 obtained from https://ladsweb.modaps.eosdis.nasa.gov and then temporally averaged

204 into the monthly LST products. In the study, the daytime LST ($LST_D$), nighttime LST

205 ($LST_N$) and the difference between daytime and nighttime LSTs ($LST_{D-N}$) at the

206 monthly scale were used.

207 The Shuttle Radar Topography Mission (SRTM) cooperated by the National

208 Geospatial Intelligence Agency (NGA) and the National Aeronautics and Space

209 Administration (NASA) provides high resolution DEMs. The SRTM DEM with the

210 spatial resolution of 90 m was downloaded from http://srtm.csi.cgiar.org/ and then

211 resampled to 1 km by the pixel averaging method. Moreover, topographical factors

212 including slope, aspect and terrain relief (Chen et al., 2020a) were extracted from the

213 SRTM DEM in ArcGIS 10.3.

214 The detailed information of the datasets used in the study is shown in Table 1.

215       Table 1 Datasets used in the study

| Data Type | Product | Spatial resolution | Temporal resolution | Source |
|---|---|---|---|---|
| Meteorological data | GPM IMERG | 10 km | Monthly | https://gpm.nasa.gov/data. |
| | Rain gauge observations | - | Daily | http://data.cma.cn/ |
| Land surface | SRTM DEM | 30 m | - | http://srtm.csi.cgiar.org/ |



| data | slope, aspect, terrain relief | 30 m | - | Derived from SRTM DEM |
|---|---|---|---|---|
| | NDVI | 1 km | Monthly | http://www.gscloud.cn/ |
| | LST | 1 km | 8-days | https://ladsweb.modaps.eosdis.nasa.gov |

**3. Methodology**
The flowchart of our method is demonstrated in Fig. 3, which includes three main
stages: (i) data processing; (ii) IMERG downscaling and (iii) downscaled IMERG
calibration. It is noted that downscaling before calibration is to avoid scale mismatch
between satellite-based areal precipitation and gauge-based point measurements.


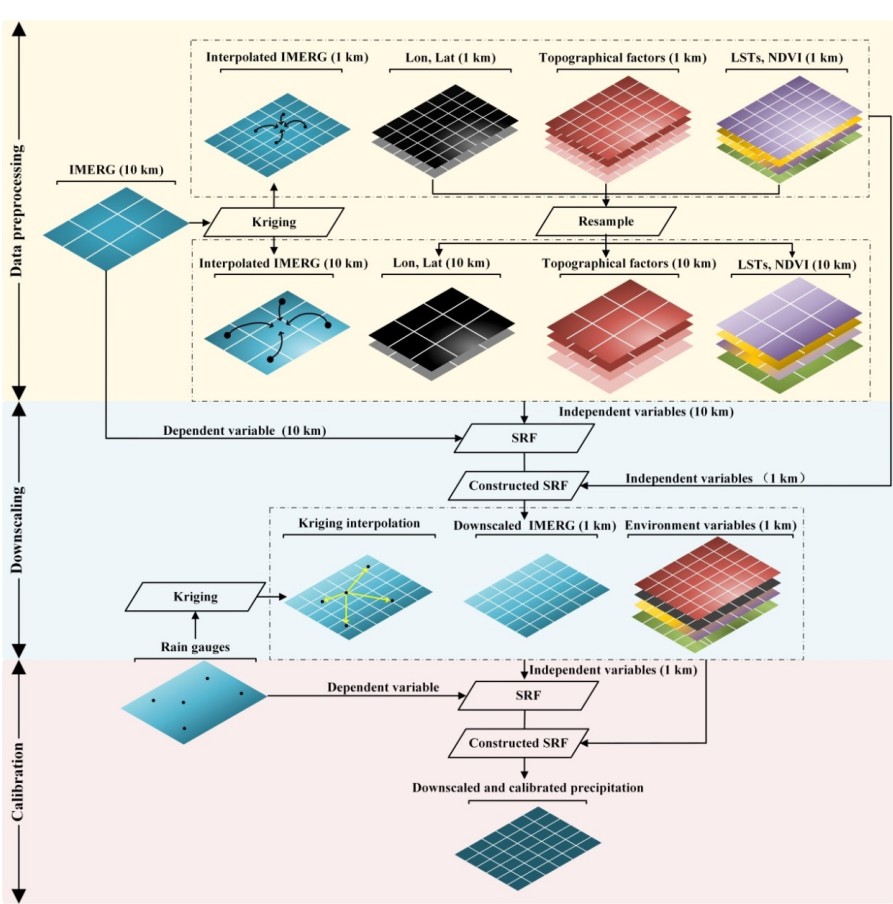

Fig. 3 Flowchart of the proposed method

### 3.1. Random Forest

RF is an ensemble of several tree predictors such that each tree relies on a random

and independent selection of features but with the same distribution (Breiman, 2001).

Specifically, each decision tree is constructed by randomly collecting some training

data with replacement while the other is used to assess the tree (sample bagging).

Moreover, while constructing each tree, only a random subset of features is selected at

each decision node (feature bagging). In the end, the majority vote for classification





or the average prediction of all trees for regression is used to obtain the final output.
Meanwhile, RF can evaluate the relative importance of the predictors by means of
out-of-bag (OOB) observations. With the OOB error, the importance of each variable
can be ranked. Many benchmarking researches have proven that RF is one promising
ML technique currently available (Hengl et al., 2018). The general framework of RF
is shown in Fig. 4.

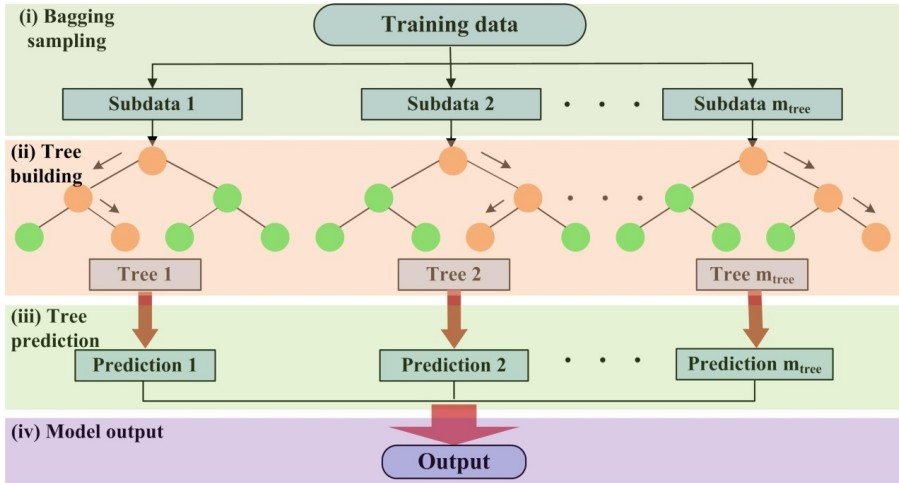


Fig. 4 General framework of RF
*3.2. Spatial Random Forest (SRF)*
In essence, the classical RF is a non-spatial statistical technique for spatial
prediction since it neglects sampling locations and general sampling pattern (Hengl et
al., 2018). This can potentially cause sub-optimal estimations, especially when the
spatial autocorrelation between dependent variables is high. To this end, a spatial RF
is proposed in this paper. The general formulation of SRF is as follows:



$$\hat{p}(s_0) = f(X_s, X_{ns}) + e$$
where $\hat{p}$ is the estimated precipitation at the location $s_0$, $e$ is the fitting residual, and $X_s$
and $X_{ns}$ are the spatial and non-spatial covariates, respectively.
In addition to spatial coordinates, one spatial covariate ($X_s$) is estimated to account
for the spatial autocorrelation between neighboring precipitation measurements, i.e.
$$X_s(s_0) = \sum_{i=1}^{n} w_i z(s_i)$$
where $z(s_i)$ is the $i$th neighboring precipitation data of the unknown point $s_0$, $w_i$ is its
weight and $n$ is the number of known data used for the estimation.
In previous studies (Li et al., 2017; Zhang et al., 2021), the inverse distance weights
(IDW) were commonly used. However, the IDW method only resorts to the spatial
distance between the estimated point and the adjacent known points, and does not
consider the spatial autocorrelation between the known points. To overcome this
limitation, the ordinary kriging-based variogram is adopted to estimate the
interpolation weights, which are obtained by solving the following linear system:
$$\begin{pmatrix} \gamma(x_1 - x_1) & \cdots & \gamma(x_1 - x_n) & 1 \\ \vdots & \ddots & \vdots & \vdots \\ \gamma(x_n - x_1) & \cdots & \gamma(x_n - x_n) & 1 \\ 1 & \cdots & 1 & 0 \end{pmatrix} \begin{pmatrix} w_1 \\ \vdots \\ w_n \\ \mu \end{pmatrix} = \begin{pmatrix} \gamma(x_1 - x_0) \\ \vdots \\ \gamma(x_n - x_0) \\ 1 \end{pmatrix}$$
where $\mu$ is Lagrange parameter and $\gamma(\cdot)$ is the semivariogram.
It can be concluded that the variogram-based weights consider the spatial
autocorrelation not only between the adjacent known points but also between the
known points and the interpolated point (Berndt and Haberlandt, 2018). Thus, it
seems more accurate than IDW. In practice, the experimental semivariogram is





estimated from sample data with the following equation (Goovaerts, 2000):
$$\hat{\gamma}(h) = \frac{1}{2n} \sum_{i=1}^{n} \left( z(\boldsymbol{x}_i) - z(\boldsymbol{x}_i + h) \right)^2$$

where $n$ is the number of data pairs with the attribute $z$ separated by distance $h$.
Generally, a theoretical semivariogram model was fitted to the experimental values
to obtain the semivariogram at any $h$. There are four commonly used theoretical
semivariogram models: the spherical, Gaussian, exponential, and power models. In
our study, the spherical model was used since it shows better results than the others in
the experiments.
**_3.3. Working procedure of the proposed method_**
The detailed steps of the proposed method are as follows (Fig. 3):
(1) Each pixel value of the 10 km IMERG was re-estimated by ordinary kriging
interpolation with its $k$ nearest neighbors (e.g. $k=8$) to obtain the interpolated
IMERG (termed as $I_s^{10km}$), the 10 km IMERG was interpolated by kriging to
obtain the interpolated 1 km IMERG ($I_s^{1km}$), and the gauge observations are
interpolated by kriging to produce the 1 km precipitation map ($P_s^{1km}$). It is noted
that the semivariogram model cannot be accurately estimated from the sparse
gauge measurements. Hence, it is difficult to accurately show the spatial
autocorrelation between the precipitation estimates. Motivated by the idea of Chen
et al. (2020c) that the satellite-based precipitation can show the spatial distribution
of precipitation, we used the satellite-based precipitation to estimate the
experimental semivariogram for interpolating gauge measurements.


(2) The negative NDVI values were excluded from the original data, which mainly

belong to snow and water bodies in the study site. The removed ones were

estimated by kriging with their neighbors, which can avoid much information loss.

(3) The 1 km environmental variables $X_{ns}^{1\text{km}}$ (i.e. NDVI, $LST_D$, $LST_N$, $LST_{D-N}$, DEM,

slope, aspect, terrain relief, latitude and longitude) were resampled to the 10 km

resolution $X_{ns}^{10\text{km}}$ by the pixel averaging method.

(4) The relationship between $X_{ns}^{10\text{km}}$, $I_s^{10\text{km}}$ and the 10 km IMERG ($IMERG^{10\text{km}}$)

was constructed by SRF:

$$IMERG^{10\text{km}}\left(s_0\right) = f_{\text{downscale}}\left(I_s^{10\text{km}}\left(s_0\right), X_{ns}^{10\text{km}}\left(s_0\right)\right) + e^{10\text{km}}\left(s_0\right)$$

where $e$ is the fitting residual.

(5) The IMERG was downscaled to 1 km ($\hat{D}^{1\text{km}}$) by the constructed relationship in

step (4) with $X_{ns}^{1\text{km}}$ and $I_s^{1\text{km}}$:

$$\hat{D}^{1\text{km}} = f_{\text{downscale}}\left(I_s^{1\text{km}}, X_{ns}^{1\text{km}}\right)$$
(6) The relationship between the 1 km predictors and the gauge observations ($G$) are

constructed by SRF:

$$G\left(s_0\right) = f_{\text{calibrate}}\left(P_s^{1\text{km}}\left(s_0\right), \hat{D}^{1\text{km}}\left(s_0\right), X_{ns}^{1\text{km}}\left(s_0\right)\right) + e^{1\text{km}}\left(s_0\right)$$
(7) The 1 km high quality precipitation data ($C^{1\text{km}}$) are produced based on the

constructed relationship in step (6):

$$C^{1\text{km}} = f_{\text{calibrate}}\left(P_s^{1\text{km}}, \hat{D}^{1\text{km}}, X_{ns}^{1\text{km}}\right)$$
In our study, residual correction was ignored during downscaling and calibration,
since many previous studies (Karbalaye Ghorbanpour et al., 2021; Lu et al., 2019)
demonstrated that residual correction on the ML-based technique decreased the



prediction accuracy.

### 3.4. Comparative methods

In the study, the performance of our method was comparatively assessed using
three manners. Firstly, we compared the results of the proposed method with those of
the classical methods including GWR, RF and BPNN. Secondly, our methodology
was compared with two classical frameworks: (i) the IMERG was downscaled by the
bilinear interpolation and then calibrated by SRF (termed as Bi-SRF), and (ii) the
IMERG was downscaled by SRF and then calibrated by GDA (termed as SRF-GDA).
Thirdly, our monthly-based estimation method was compared with the annual-based
SRF fraction disaggregation method (termed as SRFdis). Finally, the results of our
method were compared with that from ordinary kriging interpolation only on gauge
measurements (termed as kriging). Overall, the proposed method was compared with
seven classical methods in our study, including GWR, RF, BPNN, Bi-SRF, SRF-GDA,
SRFdis and kriging.
Note that the parameters of all the methods were tuned based on the trial-and-error
scheme under the $l$-fold cross validation technique (An et al., 2007). Specifically, all
gauge measurements were first divided into $l$ folds. The prediction function was
trained using $l$-1 folds, and the remainder was used for validation. The process is
repeated $l$ times until all folds were used for validation. Here, we set $l$=10. For each
group of specified parameters, the 10-fold cross validation was repeated for one time,
and the optimized parameters correspond to the minimized fitting error. Thus, the


overfitting problem could be avoided.
*3.5. Accuracy measures*
Three accuracy measures were adopted in the quantitative accuracy evaluation,
including root mean square error (RMSE), mean absolute error (MAE) and correlation
coefficient (CC) (Jing et al., 2016; Sharifi et al., 2019). They are respectively
expressed as
$$RMSE = \sqrt{\frac{1}{n}\sum_{i=1}^{n}\left(E_i - O_i\right)^2}$$

$$MAE = \frac{\sum_{i=1}^{n}\left|E_i - O_i\right|}{n}$$

$$CC = \frac{\sum_{i=1}^{n}\left(E_i - \bar{E}\right)\left(O_i - \bar{O}\right)}{\sqrt{\sum_{i=1}^{n}\left(E_i - \bar{E}\right)^2} \times \sqrt{\sum_{i=1}^{n}\left(O_i - \bar{O}\right)^2}}$$

where $n$ is the number of testing stations, $E_i$ and $O_i$ are the estimated and observed
precipitations at station $i$, respectively.
Generally, CC is used to measure the consistency between the estimated and
observed precipitations, while RMSE and MAE can assess the absolute deviation
between the estimated and observed values.
**4. Results and analysis**
We analyzed the results of the proposed method and the other methods on different
temporal scales including monthly, seasonal and annual ones, where the latter two





scales were averagely computed from the monthly one.

### 4.1. Monthly scale

Fig. 5 illustrates the scatterplots between the predicted and observed precipitations
on a monthly scale from 2015 to 2019. Results demonstrate that regardless of
accuracy measures, BPNN and GWR produce worse results than the original IMERG.
This is mainly owed to the complex relationship between the precipitation and the
predictors, which was not accurately captured by the two methods. RF performs better
than IMERG, yet worse than kriging. By contrast, the four SRF-based methods
including the proposed method, Bi-SRF, SRF-GDA and SRFdis outperform the other
methods. This reflects the significant effect of spatial autocorrelation between the
gauge measurements on capturing the complex predictors-precipitation relationship.
Moreover, the proposed method with the RMSE, MAE and CC of 33.22 mm, 19.22
mm and 0.933 produces the best result. Thus, it can be concluded that (i) SRF-based
downscaling and calibration is more effective than bilinear downscaling (Bi-SRF) and
GDA-based calibration (SRF-GDA) and (ii) there is no obvious time latency for
vegetation response to precipitation in the study site, since the proposed method is
slightly more accurate than SRFdis.

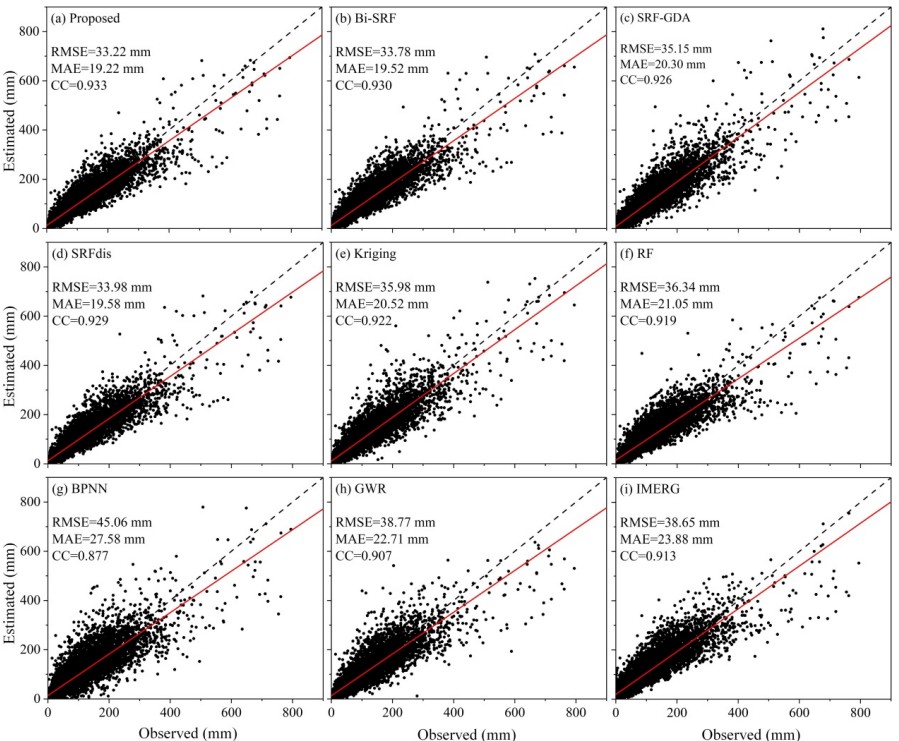


Fig. 5 Scatterplots between the estimated and the observed precipitation on a monthly

scale from 2015 to 2019



Fig. 6 shows the boxplots of the four accuracy measures. Obviously, BPNN obtains
the poorest results, with the median RMSE, MAC and CC of 30.48 mm, 22.66 mm
and 0.64, respectively. It is followed by GWR, RF and kriging. The accuracy rank is
consistent with that shown in Fig. 5. The four methods based on SRF seem more
accurate than the classical methods. SRFdis, Bi-SRF and SRF-GDA have the median
RMSEs of 21.41, 21.44 and 22.27 mm, respectively, while the proposed method has
the value of 21.03 mm. In other words, the proposed method outperforms the other
methods, which further highlights the benefit of including spatial autocorrelation
information for downscaling and calibration of satellite-based precipitation.

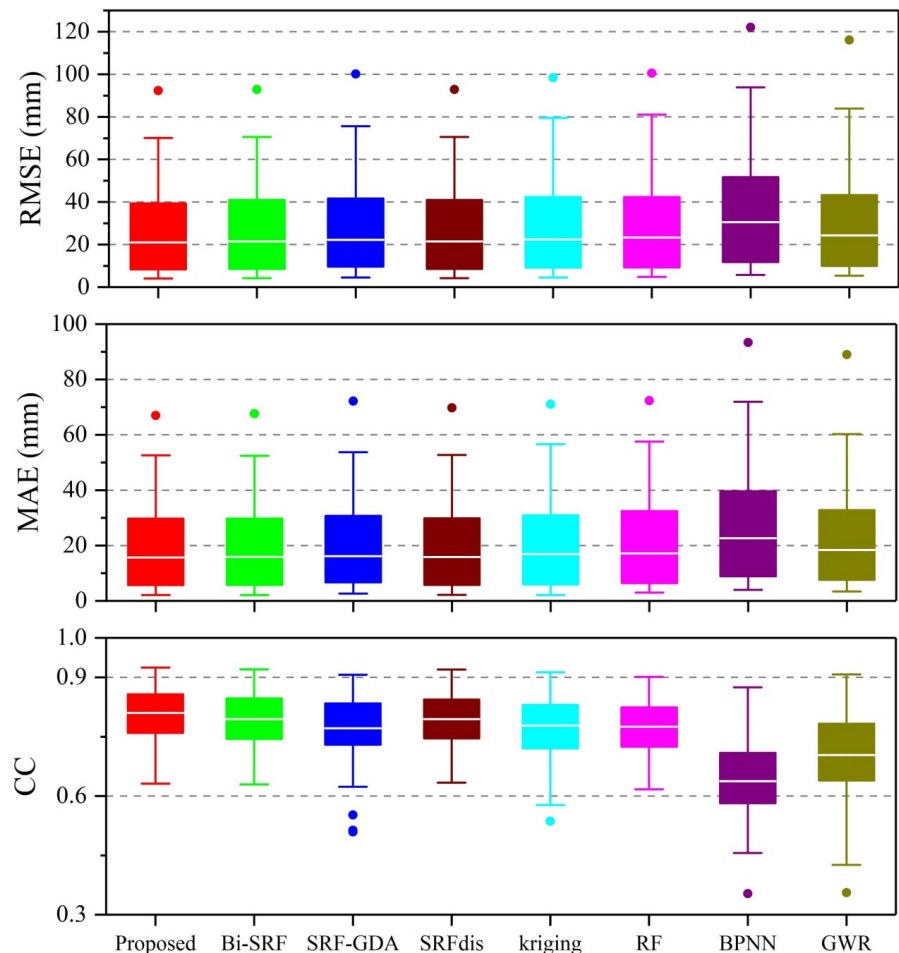


Fig. 6 Boxplots of RMSE, MAE and CC for the precipitation estimation methods on a

monthly scale during 2015-2019

Fig. 7 shows the RMSE spatial distribution of all gauge stations for the proposed
method, SRFdis, RF, BPNN, kriging and GWR. Overall, the RMSEs tend to be larger
in the middle part, since the precipitation is higher in the middle part than in the other
parts (Fig. 1). BPNN (Fig. 7d) yields the poorest results, where many stations have the
RMSEs greater than 60 mm. It is followed by GWR (Fig. 7f). RF (Fig. 7c) and
kriging (Fig. 7e) seem better than GWR and BPNN at most stations. The proposed





method (Fig. 7a) and SRFdis (Fig. 7b) are more accurate than the classical methods, especially at the stations in the middle area. Moreover, the proposed method performs better than SRFdis at some stations, such as those in the right-top.

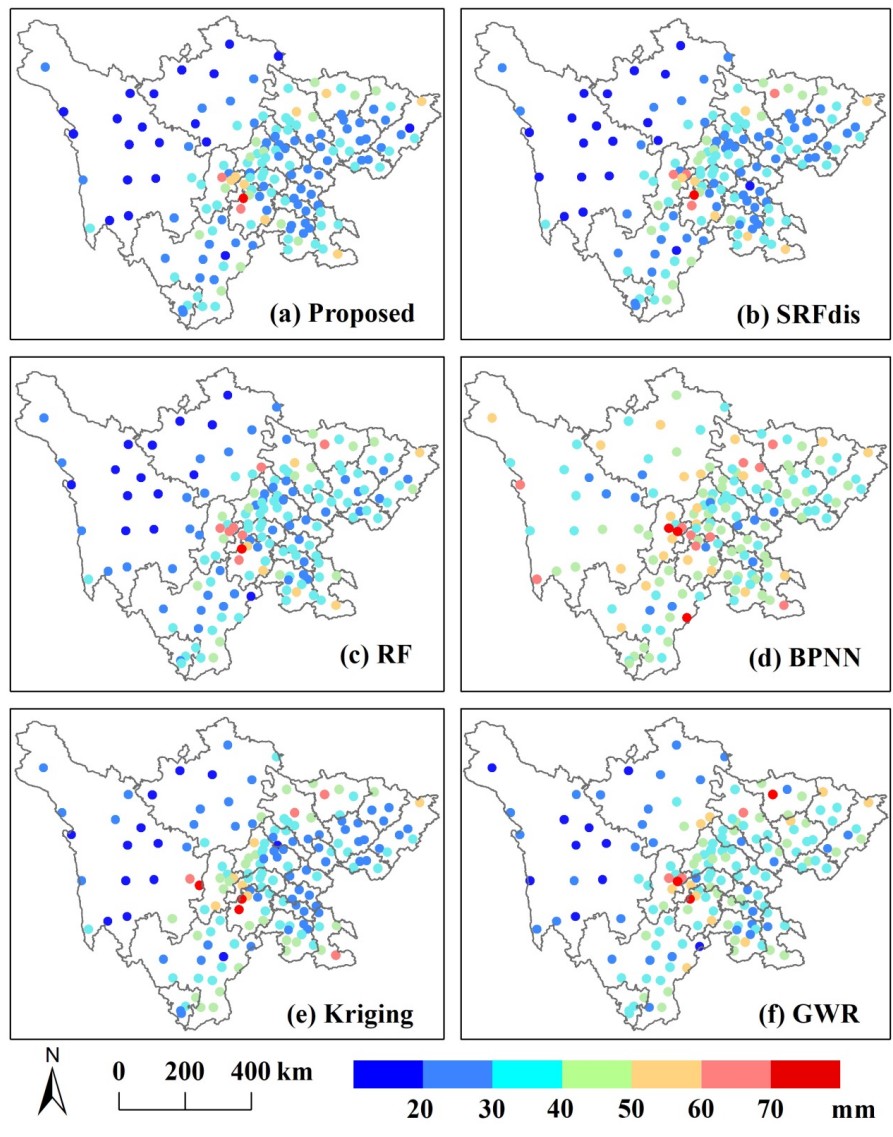

Fig. 7 RMSE distribution of all gauge stations for the proposed method and some representative methods on a monthly scale during 2015-2019



### *4.2. Seasonal scale*

The estimation errors of all the methods on a seasonal scale (i.e. spring, summer, autumn and winter) are provided in Table 2. Results indicate that regardless of accuracy measures, all methods obtain the best and the worst results in winter and in summer, respectively. This conclusion is consistent with the results yielded by (Baez-Villanueva et al., 2020; Chen et al., 2020c; Zambrano-Bigiarini et al., 2017). This could be due to the facts that (i) winter has the lowest precipitation and summer has the highest one (Fig. 2), and (ii) the large precipitation in summer was caused by complex conditions, like climatic anomaly and encounter of the cold and warm air masses, which cannot be accurately explained by the predictors (Chen et al., 2015). The accuracy rank for all the methods in the four seasons is similar. More specifically, BPNN yields worse results than IMERG in spring, summer and autumn, and a better result in winter. GWR is slightly more accurate than BPNN in the four seasons. Kriging with a similar accuracy to RF obviously outperforms BPNN and GWR. The four SRF-based methods seem more accurate than the classical methods in almost all seasons, expect for SRF-GDA in winter. Moreover, the proposed method consistently performs the best in the four seasons. Taking winter as an example, our method is about 11.44%, 8.59%, 4.77% and 2.89% more accurate than kriging, RF, BPNN and GWR, respectively.

Table 2 RMSEs, MAEs and CCs of all the estimation methods on a seasonal scale during 2015-2019 (RMSE: mm; MAE: mm)





| Season | Index | Proposed | Bi-SRF | SRF-GDA | SRFdis | Kriging | RF | BPNN | GWR | IMERG |
|--------|-------|----------|--------|---------|--------|---------|------|------|------|-------|
| | RMSE | 21.99 | 22.19 | 23.03 | 22.04 | 23.38 | 23.67 | 30.71 | 25.97 | 25.97 |
| Spring | MAE | 15.36 | 15.52 | 15.93 | 15.48 | 16.14 | 16.64 | 22.48 | 18.24 | 19.30 |
| | CC | 0.889 | 0.887 | 0.882 | 0.888 | 0.876 | 0.870 | 0.793 | 0.841 | 0.855 |
| | RMSE | 56.13 | 57.06 | 59.27 | 57.51 | 61.07 | 61.83 | 74.46 | 65.49 | 64.46 |
| Summer | MAE | 39.92 | 40.44 | 41.77 | 40.63 | 43.16 | 43.66 | 54.55 | 46.32 | 47.30 |
| | CC | 0.857 | 0.851 | 0.845 | 0.849 | 0.832 | 0.824 | 0.745 | 0.795 | 0.818 |
| | RMSE | 27.50 | 28.06 | 29.23 | 28.24 | 29.49 | 29.48 | 39.70 | 31.63 | 32.19 |
| Autumn | MAE | 17.51 | 17.89 | 18.53 | 17.96 | 18.42 | 19.25 | 26.67 | 20.79 | 21.98 |
| | CC | 0.928 | 0.925 | 0.920 | 0.924 | 0.918 | 0.917 | 0.864 | 0.902 | 0.905 |
| | RMSE | 6.29 | 6.54 | 7.70 | 6.51 | 7.01 | 6.83 | 9.29 | 8.11 | 11.28 |
| Winter | MAE | 4.11 | 4.25 | 4.97 | 4.26 | 4.36 | 4.65 | 6.64 | 5.66 | 6.93 |
| | CC | 0.853 | 0.839 | 0.790 | 0.841 | 0.823 | 0.826 | 0.688 | 0.735 | 0.595 |

To further illustrate the distributions of each accuracy measure, the boxplots of
RMSEs, MAEs and CCs in each season are provided in Figs. 8, 9 and 10, respectively.
Obviously, BPNN has the largest accuracy range in the four seasons, indicating its
instability for precipitation estimation. Moreover, it produces the largest median
RMSEs and MAEs with the values of 9.23-71.25 mm and 6.90-55.42 mm,
respectively, and the smallest median CCs with the values of 0.61-0.66. Compared to
BPNN, the RMSEs of RF and GWR are decreased to 6.90-54.92 mm and 7.04-58.17
mm, respectively, MAEs to 4.67-40.10 mm and 5.02-41.48 mm, respectively, while
CCs are increased to 0.76-0.80 and 0.39-0.73, respectively. Kriging performs better



than RF and GWR in almost all seasons, except for summer. Except for SRF-GDA,
the other SRF-based methods are more accurate than the classical methods. On the
whole, the proposed method produces the best results, with the median RMSEs,
MAEs and CCs of 6.35-52.08 mm, 4.18-38.94 mm and 0.78-0.84 in the four seasons.



Fig. 8 Boxplots of RMSEs of all the methods on the seasonal scale during 2015-2019



Fig. 9 Boxplots of MAEs of all the methods on the seasonal scale during 2015-2019



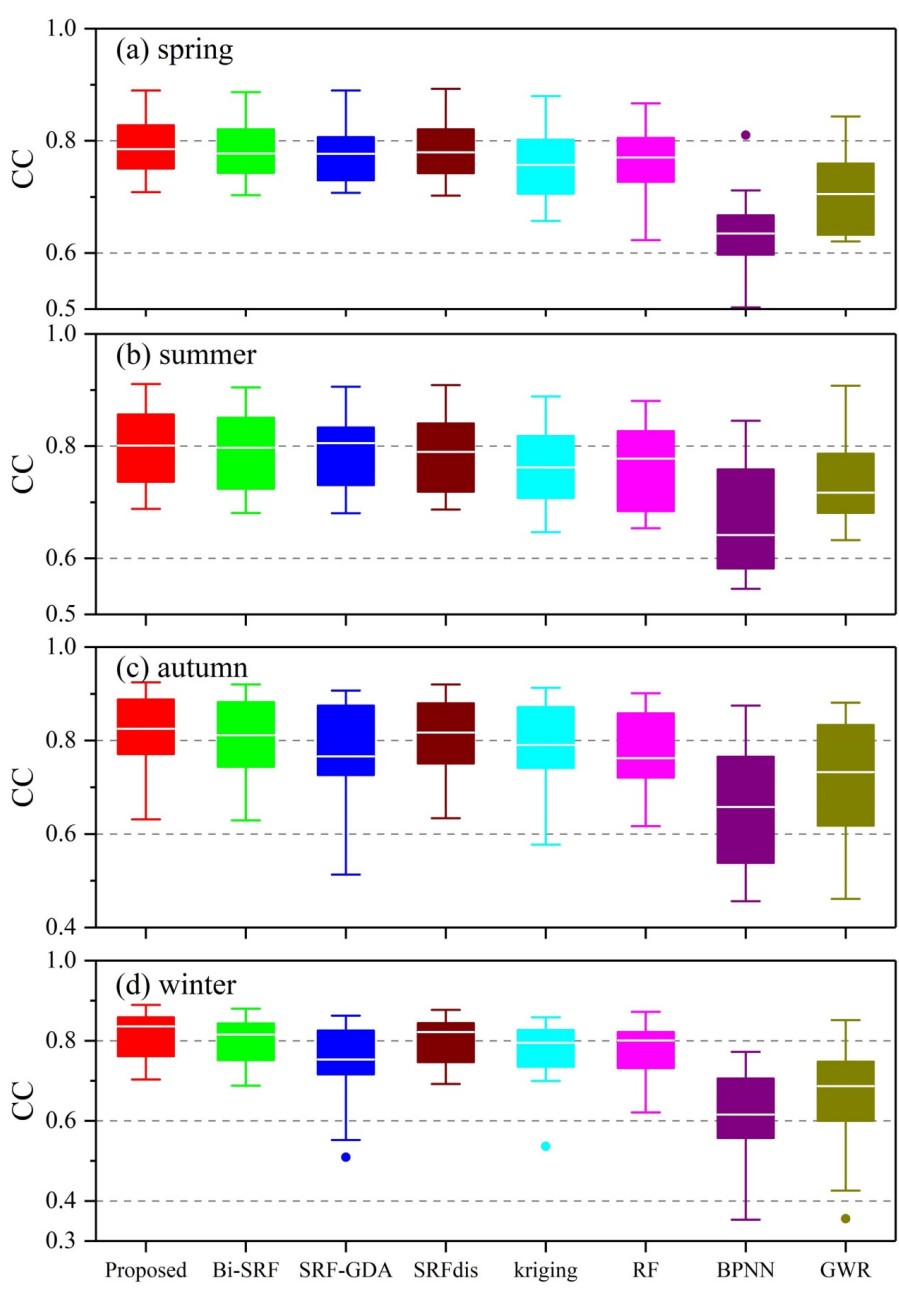


Fig. 10 Boxplots of CCs of all the methods on the seasonal scale during 2015-2019
*4.3. Annual scale*





Fig. 11 illustrates the accuracy measures of all the methods on an annual scale from
2015 to 2019. Results demonstrate that all methods produce the worst results in 2018.
This is because this year has the largest precipitation (Fig. 2). In comparison, BPNN
produces the poorest results in all years, which is followed by IMERG and GWR. RF
and kriging are consistently more accurate than BPNN, IMERG and GWR, especially
in 2017-2019. The proposed method always performs better than the other methods in
the five years, which is closely followed by Bi-SRF and SRFdis. SRF-GDA produces
worse results than the other SRF-based methods.

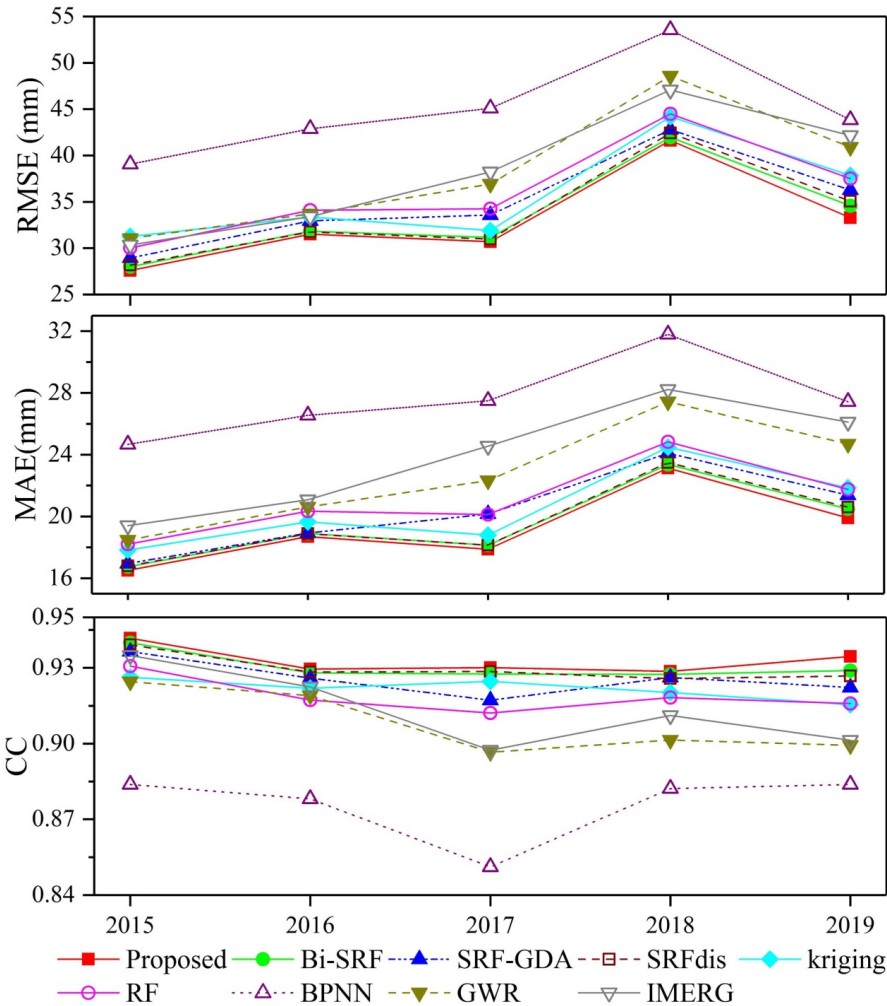


Fig. 11 Accuracy measures of all the methods on an annual scale from 2015 to 2019

Since the wettest month was July 2018 (Fig. 2), it was taken as an example to show
the precipitation estimates of the proposed method and some classical methods.
Results (Fig. 12) indicate that all the estimated precipitation maps have a similar
spatial distribution and pattern to IMERG, yet the former have more detailed
information than the latter due to the inclusion of the high-resolution predictors.
However, there exist some differences between the methods. Specifically, the kriging





map (Fig. 12b) loses many details of spatial precipitation patterns. This is expected as
it only uses ground measurements for the interpolation. RF (Fig. 12c) shows obvious
unnatural discontinuity. GWR (Fig. 12d) suffers from more variations and fractions
compared with neighbors. In comparison, the proposed method (Fig. 12a) produces a
good precipitation map.

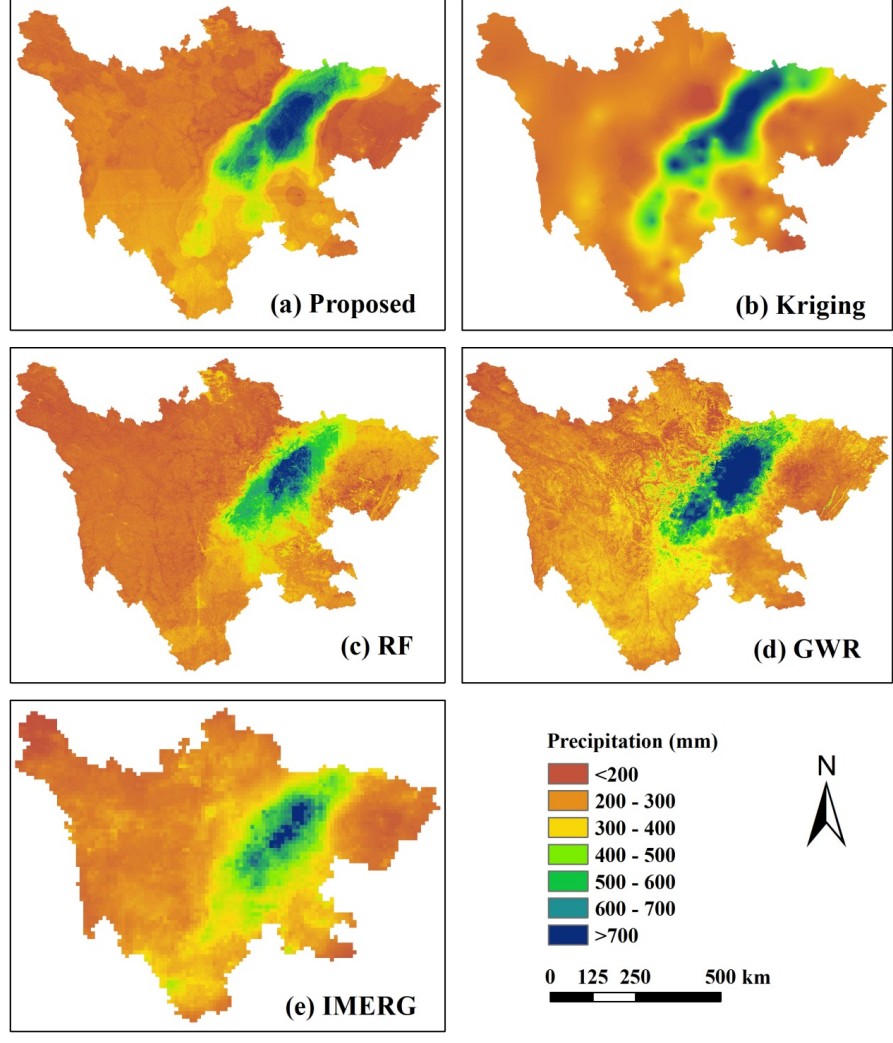


Fig. 12 Downscaled and calibrated precipitation comparison between the proposed



method and some representative methods on the wettest month

**5. Discussion**
For downscaling and calibration of satellite-based precipitation, the three most
important factors are model, predictors and temporal scale used for constructing
predictors-precipitation relationship (Chen et al., 2020b). Thus, they should be
carefully selected to produce accurate precipitation data.
*5.1. Model*
In previous studies, the most commonly adopted model is GWR (Chen et al., 2015;
Xu et al., 2015), since it has the merit of taking the spatial variation between the
predictors and precipitation into account. However, the performance of GWR
seriously depends on the density of rain gauge stations, and large interpolation errors
can be found in areas with sparse gauge stations and complex terrain characteristics
(Lu et al., 2019). Ma et al. (2017) indicated that GWR-based downscaled TRMMs
before and after residual correction for the period 2000 to 2013 at an annual scale are
less accurate than the original TRMM over the Tibet Plateau. Karbalaye Ghorbanpour
et al. (2021) showed that GWR has poorer downscaling results than the original
TRMM for 2012 and 2013 on an annual scale over Lake Urmia Basin. Our results
demonstrated that on a monthly scale (Fig. 5), GWR produces worse results than the
original IMERG, with the RMSE, MAE and CC values of 38.77 mm, 22.71 mm and
0.907, respectively. On a seasonal scale (Table 1), GWR is less accurate than IMERG



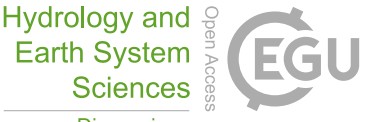

in summer, with the RMSE, MAE and CC values of 65.49 mm, 46.32 mm and 0.795,
respectively. On an annual scale (Fig. 11), compared to IMERG, the performance of
GWR is unsatisfactory in terms of CC. Moreover, the precipitation map of GWR
shows some larger values compared to their neighbors (Fig. 12d).
In contrast, the ML methods including RF and SRF are always more accurate than
GWR due to their merits for handling the complex nonlinear predictors-precipitation
relationship. This conclusion agrees well with previous studies (Karbalaye
Ghorbanpour et al., 2021; Sachindra et al., 2018). In addition, the ML methods do not
require residual correction (Jing et al., 2016; Shi et al., 2015). However, as a statistical
tool, the classical ML methods neglected the spatial autocorrelation between the
gauge measurements. Thus, a spatial RF (SRF) with the consideration of the spatial
autocorrelation information was constructed. SRF was used in both downscaling and
calibration in our study, where the original IMERG and the gauge data were
interpolated to produce input predictors for the first and second stages, respectively.
The results on the three scales demonstrated the higher accuracy of SRF than RF (see
Figs. 5-11, Table 1). Note that although kriging interpolation based on only gauge
measurements is more accurate than IMERG, BPNN and GWR, its precipitation map
is so smooth that many detailed precipitation patterns are lost (Fig. 12b).
*5.2. Environmental predictors*
NDVI, latitude, longitude and DEM-based parameters were commonly adopted
environmental variables for estimating precipitation (Shi et al., 2015). However,



satellite-based precipitation across regions with no relationship with NDVI and DEM
could not be estimated. For example, in barren or snow areas, the precipitation does
not influence NDVI due to the sparse distribution of vegetation (Xu et al., 2015).
Jing et al. (2016) indicated that the downscaled models including LST features (LSTs)
performed better those without LSTs. Thus, in addition to NDVI and DEM-related
parameters, daytime LST ($LST_D$), nighttime LST ($LST_N$), and difference between
day and night LSTs ($LST_{D-N}$) were also used in our study.

Based on RF (Breiman, 2001), the relative importance of each predictor (i.e.

predictor importance estimate) is shown in Fig. 13. Results show that precipitation
from kriging interpolation has the most importance, which indicates the significance
of the spatial autocorrelation between gauge measurements. Kriging estimation is
followed by downscaled precipitation. The three LSTs also have a great impact on
the precipitation estimation, where $LST_D$ seems more important than $LST_N$ and
$LST_{D-N}$. NDVI has a slight effect on the precipitation, which ranks last but one. This
might be due to the fact that NDVI is influenced by both precipitation and
temperature in the study site, and the low temperature above certain elevations
hinders the vegetation growth. Motivated by this idea, Wang et al. (2019) first
removed the influence of temperature on NDVI, and then used the processed NDVI
for downscaling TRMA in Qilian Mountains. Different from the aforementioned
scheme, we took both LSTs and NDVI as the predictors, and then the complex
predictors-precipitation relationship was captured by RF based on its powerful
learning ability. Among the 12 predictors, aspect has the least importance. This
conclusion was also obtained by Ma et al. (2017) for downscaling TMPA 3B43 V7
data over the Tibet Plateau. Compared to aspect, DEM and terrain slope seem more
important.

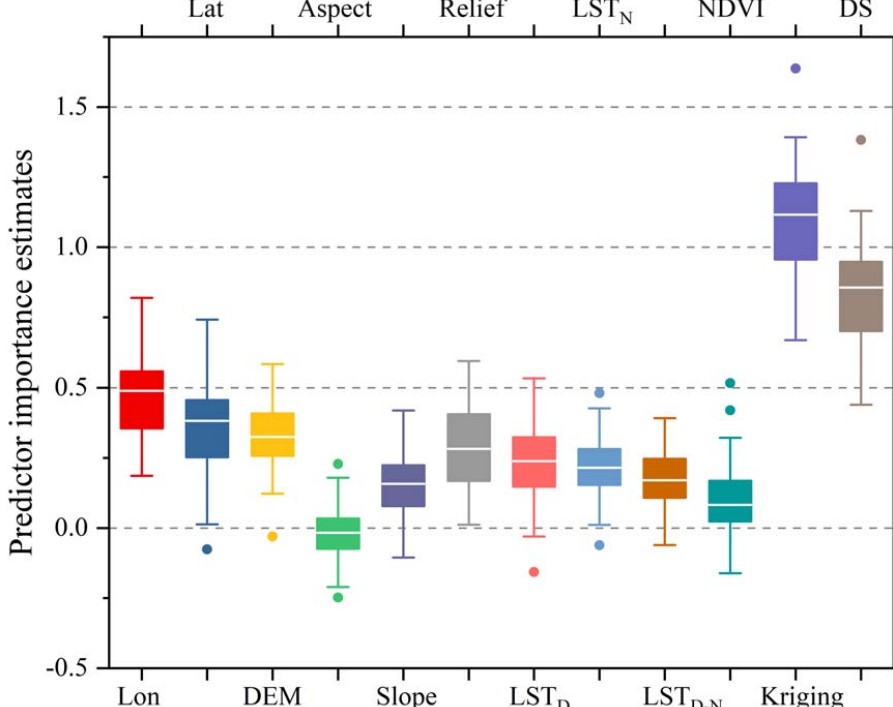


Fig. 13 Predictor importance estimates (Lat: latitude; Lon: longitude; DS: downscaled
precipitation; kriging: interpolated precipitation based kriging on gauge data)
**5.3. Temporal scale**
Temporal scale has a great effect on the selection of predictors for precipitation
estimation. There is a debate on whether NDVI should be taken as a predictor for
downscaling and calibration of monthly precipitation. Some (Duan and Bastiaanssen,
2013; Immerzeel et al., 2009) argued that NDVI cannot be used for monthly





precipitation estimation since the response of NDVI to precipitation usually delayed
for two or three months. Hence, one effective solution is to perform downscaling at
the annual scale, and then use the monthly fractions derived from the original
precipitation data to disaggregate the annual precipitation to the monthly one (i.e.
annual-based fraction disaggregation) (Duan and Bastiaanssen, 2013). However, some
(Brunsell, 2006; Chen et al., 2020c; Lu et al., 2019; Xu et al., 2015) stated that the
precipitation-NDVI relationship is hardly time-delayed, since vegetation could
influence precipitation by adjusting temperature and air moisture during the growing
seasons. Thus, it is possible to estimate precipitation with NDVI at the monthly scale.
In our study, we found that the proposed method on the monthly scale is slightly more
accurate than that on the annual scale (i.e. SRFdis) in all seasons (see Figs. 8-10),
indicating that NDVI could be used for monthly precipitation estimates in the study
site.

### *5.4. Easy-to-use feature*

Since the classical RF does not consider the spatial information in the modeling
process, Hengl et al. (2018) proposed an improved RF for spatial estimation, where
the buffer distances from the point-based measurements were taken as the predictors.
Motivated by this idea, Baez-Villanueva et al. (2020) presented a RF-based method
(RF-MEP) for merging satellite precipitation products and rain gauge measurements,
where the spatial distances from all rain gauges to the grid cells in the study site were
used as the variables. RF-MEP performed better than all precipitation products and



some merging methods. However, as stated by Baez-Villanueva et al. (2020),
RF-MEP has a huge computational cost, since the number of extra input features
equals to that of gauge measurements. Moreover, RF-MEP ignored the spatial
autocorrelation between the gauge measurements. In comparison, our SRF only
requires one extra feature that is estimated by kriging interpolation on the
precipitation measurements. Compared to the buffer distance layers, it is much more
computationally effective. Moreover, with the variogram-based kriging interpolation,
the spatial autocorrelations between the gauge measurements and between the
estimated precipitation and gauge measurements are taken into account. Thus, the
aforementioned features make our method accurate, effective and easy-to-use.
Recently, Georganos et al. (2019) proposed a geographical RF to overcome spatial
heterogeneity in remote sensing and population modelling. The geographical RF is
essentially a local interpolation method, where only the $n$ nearest observations around
the interpolated point is used. However, this kind of methods has the tendency to
produce discontinuity maps due to the local interpolation nature (Chen and Li, 2019).
Moreover, the global information inherent in the dataset cannot be used, which might
result in biased results. In comparison, our method with the aforementioned features is
highly recommended.
***5.5. Further researches***
In the further studies, we will focus on the following directions. Firstly, other land
surface variables such as soil moisture (Brocca et al., 2019; Fan et al., 2019), and



meteorological conditions such as cloud properties (Sharifi et al., 2019) could be
adopted to enhance the predictors-precipitation relationship, thereby further
improving IMERG quality. Secondly, the correction of satellite-based precipitation on
higher-temporal scales (e.g. daily or hourly) is challenging and valuable (Chen et al.,
2020b; R. Lima et al., 2021; Sun and Lan, 2021; Wu et al., 2020). Whether our
method could be applied on these scales might need validation. Thirdly, in our
experiments, all rain gauge measurements were used to improve the quality of
satellite-based precipitation. However, it is generally accepted that sample density has
a significant effect on the accuracy of the classical calibration methods
(Baez-Villanueva et al., 2020; Bai et al., 2019; Lin and Wang, 2011; Wang and Lin,
2015; Zhang et al., 2021). Thus, its influence on the results of our method should be
quantitatively assessed, thereby determining the most suitable gauge density in
different hydrological applications. Finally, numerous satellite-based precipitation
products have been available, and each one has its shortcomings and advantages for
the capture of spatial precipitation patterns (Baez-Villanueva et al., 2020; Chen et al.,
2020c). Thus, the fusion of multiple precipitation products based on our methodology
is a promising alternative to improve the quality of precipitation data. Thus, its
performance requires further assessment.
**6. Conclusions**
To enhance the resolution (from 0.1° to 1 km) and accuracy of the monthly IMERG
V06B Final Run product, a spatial RF (SRF)-based downscaling and calibration





method is proposed in this paper. The merits of the proposed method are twofold: (i)
SRF takes the spatial autocorrelation between the precipitation measurements into
account when constructing the predictors-precipitation relationship and (ii) the SRF
model is used not only in downscaling but also in calibration of IMERG, with the
incorporation of some precipitation-related high-resolution variables. The
performance of the proposed method was compared with those of seven methods
including GWR, RF, BPNN, Bi-SRF, SRF-GDA, SRFdis and kriging for enhancing
the quality and resolution of monthly IMERG across Sichuan province, China from
2015 to 2019. The main findings and conclusions can be summarized as follows:
(1) The SRF-based methods including the proposed method, Bi-SRF, SRF-GDA and

SRFdis are more accurate than the classical methods on all temporal scales.

Moreover, the proposed method ranks the first, indicating that SRF-based

downscaling and calibration is more promising than bilinear-based downscaling

and GDA-based calibration.

(2) The comparison between the monthly-based and annual-based estimation

demonstrates that there is no statistically significant difference between them,

indicating that NDVI can be used for monthly precipitation estimation in the study

site.

(3) Kriging outperforms the original IMERG, BPNN and GWR in terms of RMSE,

MAE and CC. However, its interpolation map suffers from serious loss of spatial

variation of precipitation, since it only uses the gauge measurements.

(4) Based on the variable importance assessment of RF, the precipitation interpolated



by kriging on the gauge measurements is the most important variable, whereas
terrain aspect is the least one.
Overall, the proposed methodology is general, robust, accurate and easy-to-use,
since its promising performance in the study area with an obvious heterogeneity in
terrain morphology and precipitation. Thus, it can be easily applied to other regions,
where high resolution and accurate precipitation data is urgently required.
**Data availability**
The gauge data are from the China Meteorological Data Service Center
(http://data.cma.cn, last access: January 2021). The GPM data are from
https://gpm.nasa.gov/data (last access: January 2021). The GPM data are from
http://srtm.csi.cgiar.org/ (last access: January 2021). The MOD13A3 data are from
http://www.gscloud.cn/ (last access: January 2021). The MOD11A2 data are from
https://ladsweb.modaps.eosdis.nasa.gov (last access: January 2021).
**Declaration of Competing Interest**
The authors declare that they have no known competing financial interests or
personal relationships that could have appeared to influence the work reported in this
paper.
**Author contributions**
CF and YY conceived the idea, and acquired the project and financial support. BJ
conducted the detailed analysis. CF contributed to the writing and revisions.



**Competing interests**

The authors declare that they have no conflict of interest.

**Acknowledgement**

This work was supported by the National Natural Science Foundation of China (Grant No. 41804001), Shandong Provincial Natural Science Foundation, China (Grant No. ZR2020YQ26, ZR2019MD007, ZR2019BD006), A Project of Shandong Province Higher Educational Youth Innovation Science and Technology Program (Grant No. 2019KJH007), Shandong Provincial Key Research and Development Program (Major Scientific and Technological Innovation Project) (Grant No. 2019JZZY010429) and by the Scientific Research Foundation of Shandong University of Science and Technology for Recruited Talents (Grant No. 2019RCJJ003).

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
