# Peer review of "Easy-to-use spatial Random Forest-based downscaling-calibration method for producing precipitation data with high resolution and high accuracy"

_Hydrology and Earth System Sciences, 2021_

## Referee Comment (RC3)

[referee-annotated manuscript omitted]

---

## Author Comment (AC2)

Dear Reviewer,

Thanks for your comments on our paper. Detailed comments and responses are as follows.
* * *
[0] This study used a random forest machine learning algorithm to downscale the GPM satellite precipitation measurements and calibrated them with gauge observations and the aforementioned high-resolution variables. The study is interesting and the authors presented some interesting results, while the context is hard to follow and the writing needs significant improvement. The tense of the article is very confused. The present tense and past tense are confused throughout the article, and there are also many grammatical mistakes. There are too many mistakes to point out here. It is suggested that the paper should be revised by someone who is good at English.

[Reply]

Our paper will be polished by someone who is good at English.

Major comments:

[1] You have the daily measured precipitation data, and you can also get daily GPM precipitation data. Why don't you analyze the downscaling model on the daily scale? In fact, there have been many daily downscaling models based on Random Forest. In general, the novelty of the article is not enough.

[Reply]

It is rather challenging to downscale daily precipitation data, since the relationship

between precipitation and land surface characteristics at the daily scale are more complex than at the monthly scale. Moreover, many land surface variables at the daily scale cannot be obtained. For example, we only obtained MODIS 16-day Normalized Difference Vegetation Index product (MOD13Q1) and MODIS 8-day Land Surface Temperature product (MOD10A2).

In this study, the objectives are to (i) develop an easy-to-use spatial RF (SRF) by taking into account the spatial autocorrelation between neighboring gauge measurements, and (ii) to propose a downscaling-calibration method based on SRF for producing high resolution and high accuracy precipitation data at the monthly scale. In our future work, efforts will be focused on the daily precipitation downscaling and calibration.

[2] The author did not give a clear explanation why the environment variables could be used in RF model to derive the correlations. The relationships between these variables and precipitation should be explained in detail, not just by listing the reference papers that use these variables. The correlations between each environmental variables (NDVI, LST, DEM etc.) with precipitation and their contribution to the prediction of the precipitation should be fully discussed.

[Reply]

Vegetation types have a significant impact on fluxes of sensible and latent heat into the atmosphere, obviously influencing the humidity of the lower atmosphere and further affecting moist convection (Spracklen et al., 2012). Therefore, the response of vegetation to precipitation has been widely investigated (Immerzeel et al., 2009; Wu

et al., 2019).

Precipitation can influence land surface temperature (LTS) both in the daytime and at night; rain leads to cool temperatures, and droughts often couple with heat waves (Jing et al., 2016; Trenberth and Shea, 2005).

Topography could affect the regional atmospheric circulation and the spatial pattern of precipitation through its thermal and dynamic forcing mechanisms (Jia et al., 2011; Jing et al., 2016). With the increase of elevations, the relative humidity of the air masses increases through expansion and cooling of the rising air masses, which causes precipitation (Jing et al., 2016).

The above information will be added to the revised paper.

[3] The IMERG data were fused into precipitation products from the satellite observations and gauge data. As the gauge data had been applied in the IMERG data generation (Level3), how could you reconcile the errors in SRF with the gauge data at ground used in your study?

[Reply]

The gauge data used in the Global Precipitation Climatology Centre (GPCC) rain gauge data were removed from our dataset in the study.

[4] Evaluation of the model was performed by considering rain gauge data as observations at ground. However, it assumed that the rain gauge measurements were representative values at their respective grid-cells. Although this was widely used in other literatures, the authors should discuss this issue to support their decision.

[Reply]

Each IMERG pixel represents the areal average precipitation within it, whereas rain gauge measurements are point-based. Therefore, this scale difference between point-based measurements of rain gauges and pixel-based values of the IMERG can cause errors during validation and calibration processes. One effective solution is to downscale precipitation data from coarse to fine. Here, IMERG was downscaled from 10 km to 1 km. Thus, the scale mismatch can be greatly reduced. Moreover, for each testing point, we extracted its value from the grid cells based on the bilinear interpolation on the neighbor cells to further reduce scale mismatch issue.

The above information will be added to the revised paper.

[5] This paper analyzed the accuracy of various precipitation data sets on a monthly, seasonal, and annual. There is no need to spend a lot of space in the paper to analyze the differences between seasonal and annual results, which makes the paper look more like a technical report without scientific insights. You should compare with previous literature that states if the results are in agreement or not with other studies in the field in your Discussion part. However, the present part of the discussion is rather empty, and it mainly analyzes the research results of others, and lacks the systematic analysis and discussion of the proposed method in this paper.

[Reply]

The results on the seasonal and annual scales will be removed from the revised paper.

In the discussion, the performance of the proposed method will be compared with those of the classical methods.

[6] Minor comments:

Line 158-160: "Overall, the high spatial and temporal variability of precipitation with the complex topography makes the study site ideally suitable for the evaluation of satellite-based precipitation estimates." Can you give the reasons or some references as evidences?

[Reply]

Overall, the high spatial and temporal variability of precipitation makes the study site ideal for evaluating satellite-based precipitation estimates (Karbalaye Ghorbanpour et al., 2021; Zhang et al., 2021).

The references will be added in the revised paper.

[7] Section 5.1. Please indicate the optimized hyper parameters (i.e., number of trees, depth of the tree, and number of features) for the Random Forest model.

[Reply]

This information will be added in Section 3.1 in the revised paper. It is as follows:

The general framework of RF is shown in Fig. 4, where three parameters should be optimized, i.e., number of trees, depth of the tree, and number of features.

[8] Line 269-271: "the spherical model was used since it shows better results than the others in the experiments." You should give the analysis results (data, charts, etc.) to prove the reason for choosing the spherical model.

[Reply]

The fitting results will be added in the revised paper.

[9] Line 315-316ï¼ "Our monthly-based estimation method was compared with the

annual-based SRF fraction disaggregation method (termed as SRFdis)" Please explain SRFdis in detail.

[Reply]

For SRFdis, the relationship between the precipitation and the land surface variables were constructed at the annual scale, and then the annual precipitation at 1 km for each typical year was estimated based on the constructed model. Finally, monthly fractions derived from IMERG were used to disaggregate 1 km annual precipitation to 1 km monthly precipitation.

The above information will be added to the revised paper.

[10] As can be seen from Figure 5 and Table 2, the proposed method and Bi SRF have very similar performance in most accuracy index, such as CC. However, the bilinear interpolation downscaling method is obviously easier to operate than SRF downscaling method. Is it necessary to use a more complex downscaling method to improve the CC value of 0.003?

[Reply]

As shown in Figure 5, the RMSE and MAE differences between the proposed method and Bi-SRF are 0.56 mm and 0.30 mm, respectively. We are sure that the proposed method outperforms Bi-SRF. The user can select the best one according to its requirement.

[11] line 404ï¼ Change "our method" to "the proposed method"

[Reply]

'the proposed method' will replace 'our method' in the revised paper.

[12] Line 432ï¼ Why do you say that "This is because this year has the largest precipitation (Fig. 2)." Can you explain why this is the reason for the worst results in 2018?

[Reply]

Extreme precipitation is often caused by complex environmental factors, which could result in complex predictors-precipitation relationships. Thus, the downscaling and calibration models tend to cause large estimation errors.

The above information will be added to the revised paper.

[13] Line 488: "Table 1"? Table 1 is the detailed information of the datasets used in the study.

[Reply]

It should be 'Table 2'.

[14] Line 499-500ï¼ What are the reasons that the day-night land surface temperature difference was used in this study?

[Reply]

Precipitation can influence land surface temperature (LTS) both in the daytime and at night. The day-night land surface temperature difference might be helpful to capture the complex LTS-precipitation relationship.

References

Immerzeel, W.W., Rutten, M.M., Droogers, P. (2009) Spatial downscaling of TRMM precipitation using vegetative response on the Iberian Peninsula. Remote Sensing of Environment 113 (2):362-370

Jia, S., Zhu, W., Lǜ, A., Yan, T. (2011) A statistical spatial downscaling algorithm of TRMM precipitation based on NDVI and DEM in the Qaidam Basin of China. Remote Sensing of Environment 115 (12):3069-3079

Jing, W., Yang, Y., Yue, X., Zhao, X. (2016) A Spatial Downscaling Algorithm for Satellite-Based Precipitation over the Tibetan Plateau Based on NDVI, DEM, and Land Surface Temperature. Remote Sensing 8 (8)

Karbalaye Ghorbanpour, A., Hessels, T., Moghim, S., Afshar, A. (2021) Comparison and assessment of spatial downscaling methods for enhancing the accuracy of satellite-based precipitation over Lake Urmia Basin. Journal of Hydrology 596:126055

Spracklen, D.V., Arnold, S.R., Taylor, C.M. (2012) Observations of increased tropical rainfall preceded by air passage over forests. Nature 489 (7415):282-285

Trenberth, K.E., Shea, D.J. (2005) Relationships between precipitation and surface temperature. 32 (14)

Wu, T., Feng, F., Lin, Q., Bai, H. (2019) Advanced Method to Capture the Time-Lag Effects between Annual NDVI and Precipitation Variation Using RNN in the Arid and Semi-Arid Grasslands. Water 11 (9):1789

Zhang, L., Li, X., Zheng, D., Zhang, K., Ma, Q., Zhao, Y., Ge, Y. (2021) Merging multiple satellite-based precipitation products and gauge observations using a novel double machine learning approach. Journal of Hydrology 594

---

## Author Comment (AC3)

Dear Prof. Eric Gaume,

Thanks for your comments on our paper. Detailed comments and responses are as follows.
* * *
[1] The proposed article is focused on an interesting question: the improvement of satellite-based precipitation products for the estimation of month, seasonal or annual precipitation amounts. It presents an original method aiming at improving the IMERG monthly precipitation product at local scale. The original aspect of the proposal, if compared to previously published methods consists in including spatial input variables in a random forest model: longitude, latitude and above all spatially interpolated rain gauge measurements based on ordinary kriging. The authors call therefore their method "spatial random forest".

The article is interesting and overall well written and structured, but could be improved in several ways. Moreover, it suffers from an evaluation flaw that has to be corrected to provide accurate estimates of the real performances of the tested methods and fair conclusions: i.e. the performance of the proposed method should not be evaluated on a validation set, but on a test set, totally independent from the model calibration and selection step. The confusion between validation and testing is a common error in the implementation of IA methods when cross-validation procedures are implemented for the calibration and selection of the models. This pitfall has been pointed out by numerous authors and generally leads to substantially overrate the performances of the IA models (See ref 1. and 2, hereafter). The authors should not

split theirs samples into two, but three subsample: a calibration set (a) and a validation set (b) (used for the cross-validation model adjustment procedure) but also an independent test set (c) used in the final step of model assessment. This has absolutely to be modified to my opinion to provide sensible results, before the manuscript can be published in HESS. I am not convinced that if really tested on an independent data set, the performances of the proposed method remain higher than the performances of the kriging method…

[Reply]

Based on the aforementioned comment, the following scheme will be adopted in the revised paper:

To quantitatively analyze the performance of all the methods, all rain gauge observations were randomly divided into $l$ folds (e.g. $l$=10), where the $l$-1 folds (i.e. training/validation data) was used to construct the model, while the remaining one set (i.e. testing data) to assess the performance of the model (Xu and Goodacre, 2018). During model construction, the $l$-1 folds were randomly divided into training and validation datasets with the proportions of 80% and 20%, respectively, where the former was used to train the model and the latter to validate the model (i.e. parameter optimization). Then, the model with the optimized parameters was assessed using the testing data. The aforementioned process was repeated $l$ times until all folds were taken as the testing data.

It can be found that the testing points were not used to construct the model.

[2] Some other aspects of the method and of its presentation could be improved (see

also the attached annotated manuscript):

[Reply]

   The paper will be polished according to the comments.

[3] Some implementation information is missing and could be added in the manuscript such as the nuggets and ranges of the variograms used for the spatial interpolation.

[Reply]

   Kriging was used to produce the 10 km and 1 km satellite-based precipitation products and interpolate the rain gauge observations for each month of the 5 years. Namely, 3*12*5=180 variograms were used in this study. Thus, this information cannot be listed in the paper due to the page limitation.

[4] The authors should provide the names of the software and possible libraries they have used for the implementation of RF.

[Reply]

   In the study, the RF regression model was performed with the freely available codes,                downloaded                from                the                website (https://code.google.com/archive/p/randomforest-matlab/downloads).

   The above information will be added to the revised paper.

[5] Figure 13 gives an interesting insight into the calibrated model and the driving input variables. It would be interesting, to provide an even clearer insight, to test the real added value of the input variables in the SFR model. I have the impression that the dominant variable is the kriging result and not the spatial coordinates. Could the

performances of the model based on the spatial coordinate only or the kriging result only be provided for a more complete discussion. I have the impression that removing the spatial coordinates from the input variables, as well as all other terrain characteristics will have little consequences on the model result.

[Reply]

The relative importance rank of all the variables was obtained as follows:

RF can evaluate the relative importance of the predictors by means of the out-of-bag (OOB) observations, i.e. the samples without being used for model construction. Specifically, to measure the importance of the $i$th predictor, its values are permuted while the values of the other predictors remain unchanged. Then, the OOB error based on the permuted samples is computed. Next, the importance score of the $i$th predictor is computed by averaging the difference between the OOB errors before and after the permutation. With the scores, the importance of each variable can be ranked.

It can be seen from figure 13 that except for aspect, all variables have an effect on the precipitation estimation. The dominant variable is the kriging result and not the spatial coordinates. This result is expected, since the kriging result is precipitation-related value, whereas the spatial coordinates is not directly related to precipitation.

[6] The proposed model finally mostly consists in an intelligent merging between spatially interpolated rain gauge measurements and satellite downscaled precipitation. By the way, was the downscaling step really useful (see comments in the manuscript)?

[Reply]

Downscaling can decrease the scale mismatch problem, since the original IMERG has the resolution of 10 m, while the downscaled one has the resolution of 1 km. Moreover, the 1 km IMERG show more detailed information and variation of precipitation patterns than the original one, as shown in Fig. 12.

[7] Likewise, ordinary kriging is a relatively basic interpolation approach. I wonder if co-kriging or kriging of residuals approaches, popular for spatial rainfall interpolation, could also have been tested. But this is probably not feasible for the revised version of this manuscript but a suggestion for future developments.

[Reply]

In our further researches, we will assess the performance of co-kriging or kriging of residuals approaches.

Best wishes,

Chuanfa Chen

Baojian Hu

Yanyan Li

---

## Author Comment (AC4)

Dear Reviewer,

Thanks for your comments on our paper. Detailed comments and responses are as follows.
* * *
The paper applies a machine learning technique for downscaling and calibration of precipitation based on remotely sensed inputs that also aims to incorporate the spatial structure of rainfall using spatial autocorrelation. The idea of paper is interesting and it also has a organized structure which is generally well-written. However, based on the methods applied and discussion of the results, the paper has several shortcomings that need to be addressed and further explained prior to publication.

Major comments:

[1] Several aspects of the OK based interpolated maps at 1k and 10k resolutions are not fully convincing. First, the accuracy of the OK-derived maps should be reported in order to determine reliability of the maps. Errors in the interpolated maps are going to be propagated to the errors in the spatial RF model because it is one of the covariates used, so they are important. It would be interesting to see if the large RMSE's in the middle part of the study area in fig.7 also show up with large errors or variance in the OK maps.

[Reply]

The average error maps of OK for interpolating 1 km and 10 km precipitation products will be given in the revised paper, which could give a comparison to the RMSE maps in Fig. 7.

[2] Related to this, the authors also need to further clarify the interpolation of a 1km image based on a 10km IMERG images using OK, which is a raster-to-raster interpolation performed (lines 273-284). A coarse to fine raster-based interpolation seems unusual, so that authors need to further describe this step.

[Reply]

For IMERG interpolation, the raster-based values were transformed into point-based values with the form of spatial coordinates (e.g. $x$ and $y$) and precipitation values, and then the scattered points were interpolated by OK to produce a map with the given resolution.

[3] The parameters tested and chosen for all the models, including the semi-variogram should be reported otherwise the study is not reproducible.

[Reply]

Kriging was used to produce the 10 km and 1 km satellite-based precipitation products and to interpolate the rain gauge observations for each month of the 5 years. Namely, 3*12*5=180 variograms were used in this study. The information is so much that cannot be listed in the paper due to the page limitation.

[4] It appears that the sRF model (also for the other ML techniques applied) did not include a separate testing phase . This is a standard approach applied when assessing the accuracy of a ML methods. I would suggest to also validate the models using an independent test that is not used in the training phase. Or re-configure the ML methods to split the total data into a training and a test set.

[Reply]

All the methods were assessed with separate testing points. the detailed information is as follows:

To quantitatively analyze the performance of all the methods, all rain gauge observations were randomly divided into $l$ folds (e.g. $l$=10), where the $l$-1 folds (i.e. training/validation data) was used to construct the model, while the remaining one set (i.e. testing data) to assess the performance of the model (Xu and Goodacre, 2018). During model construction, the $l$-1 folds were randomly divided into training and validation datasets with the proportions of 80% and 20%, respectively, where the former was used to train the model and the latter to validate the model for tuning parameters. Then, the model with the optimized parameters was assessed using the testing data. The aforementioned process was repeated $l$ times until all folds were taken as the testing data.

It can be found that the testing points were not used to construct the model.

[5] Discussion of the results focuses more on the positive aspects of using sRF but the authors do not give a balanced view by providing a critical analysis of the results of sRF. For instance, the accuracy metrics presented highlight that sRF performs well compared to the other models. However, visual comparison of the boxplots of these metrics alone in figs. 8-9 shows comparable accuracies all the models based on their range and median. Significant differences between the accuracies obtained, particularly in relation to sRF, should be reported to provide gravitas on the authors claim that sRF outperforms the other models.

[Reply]

In the revised paper, a balanced view will be added to the revised paper to give a fair assessment on the performance of the proposed method.

[6] There is an underestimation of precipitation values regardless of the model used based on fig 5 . This should be further elaborated in addition to the three accuracy metrics provided, so the bias of the estimates should also be reported. Furthermore, for very high precipitation values (e.g. >400mm), the scatter of the points in fig.5 becomes larger, indicating that all the models tested perform poorly at v. high rainfall amounts. It could be insightful to assess separately how the models compare for v. high rainfall conditions, since prediction of these extreme cases need to be generally improved.

[Reply]

Results illustrate that all models seem to underestimate the precipitation, especially for very high precipitation values (e.g. >400mm). This is because high precipitations are often caused by complex environmental factors, resulting in complicated predictors-precipitation relationships. Thus, more important land surface characteristics should be included into the model to improve the estimation accuracy.

The above information will be added to the revised paper.

[7] It is unclear how the importance measures are calculated from fig. 13, so this should also be included in the methodology of the paper. Furthermore, discussion of the rankings could be made more in depth by determining whether they agree or deviate (and why they do) from known controls on rainfall distribution.

[Reply]

To measure the importance of the $i$th predictor, its values are permuted while the values of the other predictors remain unchanged. Then, the OOB error based on the permuted samples is computed. Next, the importance score of the $i$th predictor is computed by averaging the difference between the OOB errors before and after the permutation. With the scores, the importance of each variable can be ranked.

Moreover, detailed analysis on the variable rank will be added in the revised paper.

[8] The authors already indicate that there is a delayed response of vegetation to rainfall. It is perhaps expected that the NDVI is one of the least important factors in the sRF model. But actually, this also provides an opportunity to also explore the lagged values of the predictors (and not only NDVI) with known delayed responses to rainfall.

[Reply]

This might be a reason for the least importance of NDVI. The above information will be added to the revised paper.

Minor comments:

[9] The captions of the figures need to be improved. Some of the features in multi-plot figures are hard to understand because of the captions are highly simplified.

[Reply]

The captions of the figures will be added to better understand in the revised paper.

[10] The final version of the manuscript will benefit for another round a English check as some sentences a phrased a bit vaguely (e.g. line 150-151)

[Reply]

Our paper will be polished by a naïve English speaker.

Best wishes,

Chuanfa Chen

Baojian Hu

Yanyan Li

---

## Author Comment (AC5)

Dear Prof. Priscilla Minotti,

Thanks for your comments on our paper. Detailed comments and responses are as follows.
* * *
The paper is well designed, nicely written, and presents a novel way to downscale precipitation data by combining semivariogram modeling into spatial random forest with well-known precipitation predictors. Although I consider the paper can be published "as is", I have suggestions for the authors:

[1] The proposed method could be given a name.

[Reply 1]

The proposed method is an easy-to-use spatial Random Forest-based downscaling-calibration method. Thus, it will be termed as SRF-DC in the revised paper.

[2] In 5.5. Further Research, the authors could also address some of the following:

[2.1] the upscaling from 1 km to the 10 km IMERG grid was done by pixel averaging. If some other aggregation stats were used (eg. median, max, mode), would the performance of some of the environmental predictors improve (eg. aspect)?

[Reply 2.1]

After our analyses, we found that the pixel averaging is slightly more accurate than the other operations. This is because the average value reflects the overall trend within each 10 km pixel and reduces the influence of outliers in the 1 km pixels (Karbalaye Ghorbanpour et al., 2021).

The above information will be added to the revised paper.

 [2.2] Some other predictors could also be included, such as position based on metric distances instead of latitude-longitude, EVI (with is better related to water content in plants or soil than NDVI), wind or atmospheric pressure features.

[Reply 2.2]

   --The predictor of position based on metric distances instead of latitude-longitude was not used in our method. The reasons are as follows:

"Since the classical RF does not consider the spatial information in the modeling process, Hengl et al. (2018) proposed an improved RF for spatial estimation, where the buffer distances from the point-based measurements were taken as the predictors. Motivated by this idea, Baez-Villanueva et al. (2020) presented a RF-based method (RF-MEP) for merging satellite precipitation products and rain gauge measurements, where the spatial distances from all rain gauges to the grid cells in the study site were used as the variables. RF-MEP performed better than all precipitation products and some merging methods. However, as stated by Baez-Villanueva et al. (2020), RF-MEP has a huge computational cost, since the number of extra input features equals to that of gauge measurements. Moreover, RF-MEP ignored the spatial autocorrelation between the gauge measurements. In comparison, SRF-DC only requires one extra feature that is estimated by kriging interpolation on the precipitation measurements. Compared to the buffer distance layers, it is much more computationally effective. Moreover, with the variogram-based kriging interpolation, the spatial autocorrelations between the gauge measurements and between the

estimated precipitation and gauge measurements are taken into account. Thus, the aforementioned features make our method accurate, effective and easy-to-use."

 The above information can be found in the Discussion.

 --We compared the performance of NDVI-based (proposed method) and EVI-based SRF (SRFEVI), and found that the former is slightly more accurate than the latter (Fig. 1). Thus, the NDVI was used in the study site.

[Figure]

Figure 1 Performance comparison between NDVI-based and EVI-based SRF (i.e. proposed and SRFEVI methods)

 --The predictors including wind or atmospheric pressure features were not used in our method, since the highest resolution of the publicly available dataset was 0.1°. The environmental factors should have the resolution of 1 km, since we aim to downscale IMERG from 0.1° to 1 km, and the 1 km environmental factors are taken as the input for the trained SRF.

[2.3] The proposed method could be transferred to use with CHIRPS or TRMM data in other parts of the world, particularly in large tracts of South America, which have complex topography and sparse gauging stations.

[Reply]

 Yes. As stated in the last paragraph of our method, "Overall, the proposed

methodology is general, robust, accurate and easy-to-use, since its promising performance in the study area with an obvious heterogeneity in terrain morphology and precipitation. Thus, it can be easily applied to other regions, where high resolution and accurate precipitation data is urgently required."

**References**

Baez-Villanueva, O.M., Zambrano-Bigiarini, M., Beck, H.E., McNamara, I., Ribbe, L., Nauditt, A., Birkel, C., Verbist, K., Giraldo-Osorio, J.D., Xuan Thinh, N. (2020) RF-MEP: A novel Random Forest method for merging gridded precipitation products and ground-based measurements. Remote Sensing of Environment 239:111606

Hengl, T., Nussbaum, M., Wright, M.N., Heuvelink, G.B., Gräler, B.J.P. (2018) Random forest as a generic framework for predictive modeling of spatial and spatio-temporal variables. PeerJ 6:e5518

Karbalaye Ghorbanpour, A., Hessels, T., Moghim, S., Afshar, A. (2021) Comparison and assessment of spatial downscaling methods for enhancing the accuracy of satellite-based precipitation over Lake Urmia Basin. Journal of Hydrology 596:126055

Best wishes,

Chuanfa Chen

Baojian Hu

Yanyan Li

---

## Author Comment (AC6)

Dear Reviewer,

Thanks for your comments on our paper. Detailed comments and responses are as follows.
* * *
The paper applies a machine learning technique for downscaling and calibration of precipitation based on remotely sensed inputs that also aims to incorporate the spatial structure of rainfall using spatial autocorrelation. The idea of paper is interesting and it also has a organized structure which is generally well-written. However, based on the methods applied and discussion of the results, the paper has several shortcomings that need to be addressed and further explained prior to publication.

Major comments:

[1] Several aspects of the OK based interpolated maps at 1k and 10k resolutions are not fully convincing. First, the accuracy of the OK-derived maps should be reported in order to determine reliability of the maps. Errors in the interpolated maps are going to be propagated to the errors in the spatial RF model because it is one of the covariates used, so they are important. It would be interesting to see if the large RMSE's in the middle part of the study area in fig.7 also show up with large errors or variance in the OK maps.

[Reply]

Since the wettest month is July 2018 (Fig. 2), it is taken as an example to show the variance of the prediction errors of OK. The prediction error map derived from Eq. (4) shows that the errors in the west are larger than in the east, and in the boundary are

larger than in the inner. It can be inferred that large errors are mainly located in the areas with the sparse distribution of rain gauges, which are not related to the RMSE distribution (Fig. 7) and precipitation (Fig. 8).

[Figure]

(b) Prediction error map

Fig. 9 Semivariogram and prediction error map of kriging on the wettest month (July 2018)

The above information will be shown in the revised paper.

[2] Related to this, the authors also need to further clarify the interpolation of a 1km image based on a 10km IMERG images using OK, which is a raster-to-raster interpolation performed (lines 273-284). A coarse to fine raster-based interpolation seems unusual, so that authors need to further describe this step.

[Reply]

For IMERG interpolation, the raster-based IMMERG were first transformed into point-based form with spatial coordinates (e.g. $x$ and $y$) and precipitation values, and then the scattered points were interpolated by OK to produce a map with the given resolution.

The above information will be added to the revised paper.

[3] The parameters tested and chosen for all the models, including the semi-variogram should be reported otherwise the study is not reproducible.

[Reply]

Kriging was used to produce the 10 km and 1 km satellite-based precipitation products and to interpolate the rain gauge observations for each month of the 5 years. Namely, 3*12*5=180 variograms were used in this study. Similarly, the other methods require at least 12*5=60 groups of parameters. The information is so much that cannot be listed in the paper due to the page limitation.

Since the wettest month is July 2018 (Fig. 2), it is taken as an example to show the semivariogram of OK. For OK, the semivariogram and its prediction error map are shown in Fig. 9. It can be found that it is a spherical model with the nugget variance (C0) of 10.0 m$^2$ , sill (C0+C) of 10,560 m$^2$, residual sum of squares (Rss) of 8,800,611 m$^2$, range (A0) of 321,000 m, and fitting R$^2$ of 0.962, respectively.

[Figure]

Semivariogram of OK

[4] It appears that the sRF model (also for the other ML techniques applied) did not include a separate testing phase . This is a standard approach applied when assessing the accuracy of a ML methods. I would suggest to also validate the models using an independent test that is not used in the training phase. Or re-configure the ML methods to split the total data into a training and a test set.

[Reply]

All the methods were assessed with separate testing points. The detailed information is as follows:

To quantitatively analyze the performance of all the methods, all rain gauge observations were randomly divided into $l$ folds (e.g. $l$=10), where the $l$-1 folds (i.e. training/validation data) was used to construct the model, while the remaining one set

(i.e. testing data) to assess the performance of the model (Xu and Goodacre, 2018). During model construction, the $l$-1 folds were randomly divided into training and validation datasets with the proportions of 80% and 20%, respectively, where the former was used to train the model and the latter to validate the model for tuning parameters. Then, the model with the optimized parameters was assessed using the testing data. The aforementioned process was repeated $l$ times until all folds were taken as the testing data.

It can be found that the testing points were not used to construct the model.

[5] Discussion of the results focuses more on the positive aspects of using sRF but the authors do not give a balanced view by providing a critical analysis of the results of sRF. For instance, the accuracy metrics presented highlight that sRF performs well compared to the other models. However, visual comparison of the boxplots of these metrics alone in figs. 8-9 shows comparable accuracies all the models based on their range and median. Significant differences between the accuracies obtained, particularly in relation to sRF, should be reported to provide gravitas on the authors claim that sRF outperforms the other models.

[Reply]

In the revised paper, a balanced view will be added to the revised paper to give a fair assessment on the performance of the proposed method. The information is as follows:

Although SRF-DC shows promising results than the classical methods, it still suffers from some limitations, which should be solved in the further researches.

Firstly, SRF-DC is more complex than Bi-SRF and SRF-GDA, since SRF is used in both downscaling and calibration in SRF-DC. Hence, applying SRF to downscale IMMERG might not be prerequisite since SRF-DC is only slightly better than Bi-SRF. However, SRF should be used to calibrate IMMERG due to the obviously higher accuracy of SRF-DC than SRF-GDA.

Secondly, SRF-DC has an obvious underestimation on high precipitation values mainly due to the omission of some important land surface variables for precipitation estimation. Thus, other available variables such as soil moisture (Fan et al., 2019; Brocca et al., 2019), and meteorological conditions such as cloud properties (Sharifi et al., 2019) should be adopted to further improve IMERG quality.

Thirdly, the correction of satellite-based precipitation on higher-temporal scales (e.g. daily or hourly) is challenging and valuable (Wu et al., 2020; Chen et al., 2020b; R. Lima et al., 2021; Sun and Lan, 2021). Whether SRF-DC could be applied on these scales requires further validation.

Finally, numerous satellite-based precipitation products have been available, and each one has its shortcomings and advantages for the capture of spatial precipitation patterns (Chen et al., 2020c; Baez-Villanueva et al., 2020). Thus, the fusion of multiple precipitation products based on SRF-DC is a promising alternative to improve the quality of precipitation data.

[6] There is an underestimation of precipitation values regardless of the model used based on fig 5. This should be further elaborated in addition to the three accuracy metrics provided, so the bias of the estimates should also be reported. Furthermore,

for very high precipitation values (e.g. >400mm), the scatter of the points in fig.5 becomes larger, indicating that all the models tested perform poorly at v. high rainfall amounts. It could be insightful to assess separately how the models compare for v. high rainfall conditions, since prediction of these extreme cases need to be generally improved.

[Reply]

We will report the bias of the estimates with respect to mean error (ME). Thus, the ME will be added to the scatterplot in the revised paper.

Moreover, we compare the performance of all the methods on the very high precipitation values (e.g. >400mm). To quantitatively analyze the performance of all methods on the observed values greater than 400 mm, their accuracy measures are shown in Table 2. Results illustrate that all methods have poor results for these observations. A possible reason is that high precipitations are often caused by complicated environmental factors, which cannot be sufficiently explained by the constructed predictors-precipitation relationship. In terms of ME, SRF-GDA ranks the first, which is followed by kriging and SRF-DC. However, their ME values are less than -70 mm. With respect to RMSE and MAE, kriging performs the best, which is closely followed by SRF-DC, and with respect to CC, SRF-DC with the value of 0.64 outperforms the others. Overall, considering the poor performance of kriging for mapping spatial precipitation distribution, SRF-DC seems the best choice for the extreme precipitation estimation.

Table 2 Accuracy measures of all methods for the observed values greater than 400

| | mm | | | |
|---|---|---|---|---|
| Method | ME (mm) | RMSE (mm) | MAE (mm) | CC |
| SRF-DC | -105.54 | 149.80 | 124.82 | 0.64 |
| Bi-SRF | -110.96 | 156.81 | 130.67 | 0.60 |
| SRF-GDA | -74.21 | 150.10 | 126.02 | 0.55 |
| SRFdis | -117.31 | 160.11 | 137.29 | 0.61 |
| Kriging | -86.25 | 146.94 | 119.53 | 0.58 |
| RF | -141.53 | 177.71 | 150.83 | 0.61 |
| BPNN | -118.88 | 171.23 | 142.00 | 0.57 |
| GWR | -139.02 | 178.85 | 145.19 | 0.57 |
| IMERG | -136.22 | 173.24 | 143.69 | 0.55 |

The above information will be shown in the revised paper.

[7] It is unclear how the importance measures are calculated from fig. 13, so this should also be included in the methodology of the paper. Furthermore, discussion of the rankings could be made more in depth by determining whether they agree or deviate (and why they do) from known controls on rainfall distribution.

[Reply]

To measure the importance of the $i$th predictor, its values are permuted while the values of the other predictors remain unchanged. Then, the OOB error based on the permuted samples is computed. Next, the importance score of the $i$th predictor is computed by averaging the difference between the OOB errors before and after the permutation. With the scores, the importance of each variable can be ranked.

Based on RF, the relative importance of each predictor (i.e. predictor importance estimate) is shown in Fig. 10. Results show that precipitation from kriging interpolation has the most importance. This is because the interpolated value is directly related to precipitation. Kriging estimation is followed by the downscaled precipitation. Longitude is the third most important variable, which is followed by latitude. This result is consistent with that of Karbalaye Ghorbanpour et al. (2021). They indicated that compared to NDVI, LST and DEM, longitude ranks the first with respect to importance score.

The three LSTs also have a great impact on the precipitation estimation, where $LST_D$ seems slightly more important than $LST_N$ and $LST_{D-N}$. NDVI has a slight effect on the precipitation, which ranks last but one. This might be due to the fact that NDVI is influenced by both precipitation and temperature in the study site, and the low temperature above certain elevations hinders the vegetation growth. It should be noted that it is less likely that the response of vegetation to precipitation has the delay in the study site, since SRF-DC on the monthly scale is more accurate than SRFdis on the annual scale.

Among the 12 predictors, aspect has the least importance. This conclusion was also obtained by Ma et al. (2017) for downscaling TMPA 3B43 V7 data over the Tibet Plateau. Compared to aspect, DEM, terrain relief and slope seem more important, since precipitation shows obvious relationships with topography. This is consistent with previous studies (Immerzeel et al., 2009; Jing et al., 2016).

The above information will be shown in the revised paper.

[8] The authors already indicate that there is a delayed response of vegetation to rainfall. It is perhaps expected that the NDVI is one of the least important factors in the sRF model. But actually, this also provides an opportunity to also explore the lagged values of the predictors (and not only NDVI) with known delayed responses to rainfall.

[Reply]

This might be a reason for the least importance of NDVI. However, in this study, it is found that SRF-DC on the monthly scale is slightly more accurate than that on the annual scale (i.e. SRFdis), indicating that the response of vegetation to precipitation has no obvious time delay.

Minor comments:

[9] The captions of the figures need to be improved. Some of the features in multi-plot figures are hard to understand because of the captions are highly simplified.

[Reply]

The captions of the figure will be added to better understand in the revised paper. It is as follows:

[Figure]

[10] The final version of the manuscript will benefit for another round a English check as some sentences a phrased a bit vaguely (e.g. line 150-151)

[Reply]

Our paper will be polished by a naïve English speaker.

Best wishes,

Chuanfa Chen

Baojian Hu

Yanyan Li

---

## Author Comment (AC7)

Dear Prof. Carla Ferreira,

Thanks for your comments on our paper. Detailed comments and responses are as follows.
* * *
Thank you very much for submitting your work to HESS. The manuscript presents a new machine learning method to downscale precipitation from satellite data. The topic is very interesting and fits well in the content of the journal, however, major revisions must be performed before publication.

[1] Please, consider all the comments suggested by the four reviewers. The comments are highly pertinent and relevant to improve the quality of your manuscript.

[Reply]

 We will give detailed responses to the comments suggested by the four reviewers, and revisions will be performed on our paper based on the comments.

[2] Additionally, I suggest to improve the Discussion section, by better highlighting the advantages and limitation of the proposed method, including the outscalling potential to other areas/study sites, and the possibility/limitations of the methodology for other time scales (e.g. daily).

[Reply]

 Based on the comments of the four reviewers, the limitations of the proposed method will be discussed and its potential will be shown in the revised paper.

[3] In Fig. 2, please correct the "Year-Month" format within the numbers in the xx axis. In Fig. 5, please specify the meaning of the red line. In Fig. 12, please specify in

the legend, within brackets, the wettest month (month and year) considered.

[Reply]

Our paper will be revised based on above mentioned comments.

[4] Authors are responsible for preparing their papers in correct English language, so please check and improve language editing.

[Reply]

Our paper will be fully polished.

Looking forward of the revised version of your manuscript!

Best wishes,

Chuanfa Chen

Baojian HU

Yanyan Li